

# Vertical distribution of chlorophyll in dynamically distinct regions of the southern Bay of Bengal

Venugopal Thushara[1], Puthenveettil Narayana Menon Vinayachandran[1], Adrian J. Matthews[2], Benjamin G. M. Webber[3], and Bastien Y. Queste[3]

[1]Centre for Atmospheric and Oceanic Sciences, Indian Institute of Science, Bangalore, India.
[2]Centre for Ocean and Atmospheric Sciences, School of Environmental Sciences and School of Mathematics, University of East Anglia, Norwich, UK
[3]Centre for Ocean and Atmospheric Sciences, School of Environmental Sciences, University of East Anglia, Norwich, UK

*Correspondence to:* P. N. Vinayachandran (vinay@iisc.ac.in)

**Abstract.**

The Bay of Bengal (BoB) generally exhibits surface oligotrophy, due to nutrient limitation induced by strong salinity stratification. Nevertheless, there are hot spots of biological activity in the BoB where the monsoonal forcings are strong enough to break the stratification; one such region being the southern BoB, east of Sri Lanka. A recent field program conducted during the summer monsoon of 2016, as a part of the Bay of Bengal Boundary Layer Experiment (BoBBLE), provides a unique high-resolution dataset of the vertical distribution of chlorophyll in the southern BoB using ocean gliders along with shipboard CTD measurements. Observations were carried out for a duration of 12–20 days during a suppressed phase of the Boreal Summer Intraseasonal Oscillation (BSISO), along a longitudinal transect at 8° N, extending from 85.3–89° E, covering the dynamically active regions of the Sri Lanka Dome (SLD) and the Southwest Monsoon Current (SMC). Mixing and upwelling induced by the monsoonal wind forcing enhanced chlorophyll concentrations (0.3–0.7 mg m$^{-3}$) in the surface layers. Observations reveal the presence of prominent deep chlorophyll maxima (DCM; 0.3–1.2 mg m$^{-3}$) at intermediate depths (20–50 m), generally below the mixed layer and above the thermocline, signifying the contribution of subsurface productivity on the biological carbon cycling in the BoB. The shape of chlorophyll profiles varied in different dynamical





regimes indicating that the mechanisms determining the vertical distribution of chlorophyll are intricate; upwelling favoured sharp and intense DCM, whereas mixing resulted in diffuse and weaker DCM. Within the SLD, open ocean Ekman pumping and the doming of thermocline favoured a substantial increase in chlorophyll concentration. Farther east, the thermocline was deeper and moderate surface blooms were

5 triggered by intermittent mixing events. Stabilising surface freshening events and barrier layer formation were often observed to inhibit the surface blooms. The pathway of SMC intrusion was marked by a distinct band of chlorophyll, indicating the advective effect of biologically rich Arabian Sea waters. The region of monsoon current exhibits the strongest DCM as well as the highest column-integrated chlorophyll. Observations suggest that the persistence of DCM in the southern BoB is promoted by surface

oligotrophy, which reduces the self-shading effect of phytoplankton and shallow mixed layers, which prevent the vertical redistribution of subsurface phytoplankton. Results from a coupled physical-ecosystem model substantiate the dominant role of mixed layer processes associated with the monsoon in controlling the nutrient distribution and biological productivity in the southern BoB. The present study provides new insights into the vertical distribution of chlorophyll in the BoB, which is not captured in satellite mea-

surements, emphasizing the need for extensive in situ sampling and ecosystem model-based efforts for a better understanding of the monsoonal bio-physical interactions and the potential climatic feedbacks.



## 1  Introduction

The Bay of Bengal (BoB) is fascinating with its unique upper ocean features strongly linked to the Indian Summer Monsoon (ISM) variability (Gadgil et al., 1984; Vecchi and Harrison, 2002; Shankar et al., 2007). The upper layer of the BoB, especially the northern BoB, is highly stable, owing to strong near-surface salinity stratification in the presence of abundant freshwater influx from precipitation and rivers. The low salinity cap in the surface layers of the BoB leads to the formation of a shallow mixed layer and a barrier layer beneath (Vinayachandran et al., 2002; Wijesekera et al., 2016a), controlling air-sea interactions and the upper ocean heat budget (Shenoi et al., 2002). In addition, monsoonal winds are relatively weak over the BoB, leading to a sluggish upper ocean, where vertical overturning and mixing processes are weak (Shetye et al., 1991; Madhupratap et al., 1996; Kumar et al., 2002; McCreary et al., 2009; Wiggert et al., 2009). Hence, destabilisation of salinity stratification is difficult, controlled by the competing effects of winds and freshwater influx. This dynamical set up imparts strong nutrient limitation on phytoplankton growth, leading to weak biological productivity in the BoB (Gomes et al., 2000; Kumar et al., 2002; Madhupratap et al., 2003). Compared to the highly productive Arabian sea, chlorophyll distribution in the BoB is often light limited, despite being located in the same tropical band, due to large cloud cover during the active phase of the monsoon (Kumar et al., 2010). In addition, the presence of suspended sediments in the vicinity of discharge from major rivers reduces the light availability for photosynthesis (Gomes et al., 2000; Kumar et al., 2004).

Though the basin averaged productivity is weak in the BoB, satellite and in situ observations reveal the presence of intense regional blooms (Vinayachandran and Mathew, 2003; Kumar et al., 2004; Kumar et al., 2007). In the northern BoB where stratification is strong, surface chlorophyll blooms are

rarely observed, except those associated with coastal processes and eddy activity. The northwestern BoB

is characterised by seasonal blooms in the presence of strong coastal upwelling induced by the alongshore

winds during the summer monsoon (Shetye et al., 1991), which enriches the previously nutrient-limited

euphotic zone (Thushara and Vinayachandran, 2016). In addition, nutrients supplied through the mon-

soonal river discharge support intense bloom activity in the nearby coastal oceans (Kumar et al., 2004;

Kumar et al., 2007). The occurrence of mesoscale eddies is an additional forcing, favouring biological

productivity through the vertical supply of nutrients (Kumar et al., 2007; Nuncio and Kumar, 2013). Pro-

ductivity in the BoB is mostly confined to the coastal ocean and dynamical regions of the open ocean, such

as the southern BoB, where the freshwater effects are relatively weaker (Vinayachandran and Mathew,

2003).

The southern BoB, characterised by strong currents, intense mixing and upwelling, is one of the most

dynamically active regions of the northern Indian Ocean (Murty et al., 1992; Schott et al., 1994; McCreary et al.,

1996; Vinayachandran and Yamagata, 1998; Vinayachandran et al., 1999; Shankar et al., 2002; Lee et al.,

2016; Wijesekera et al., 2016b). Unlike the northern BoB, salinity stratification is relatively weak in

the south, resulting in a deeper mixed layer. Prominent chlorophyll blooms are observed in the coastal

and open ocean regions of the southern BoB, closely linked to monsoon circulation (Vinayachandran,

2009). The region off the south coast of Sri Lanka is characterised by intense summer blooms trig-

gerred by the coastal upwelling of nutrients (Vinayachandran et al., 2004). Cyclonic wind stress curl

east of Sri Lanka during the summer monsoon leads to the formation of the Sri Lanka Dome (SLD;

Vinayachandran and Yamagata, 1998), where open ocean Ekman pumping of nutrients triggers bloom

generation (Vinayachandran et al., 2004). The Southwest Monsoon Current (SMC) intruding into the



southern BoB (Vinayachandran and Yamagata, 1998; Vinayachandran et al., 2013; Jensen, 2001) carries biologically rich waters from the Indian and Sri Lankan coasts, supporting bloom activity all along its path (Vinayachandran et al., 2004). After finding its way into the BoB, the SMC bifurcates into several branches and the associated cold-core eddies are observed to enhance chlorophyll concentrations (Jyothibabu et al., 2015). During the winter monsoon, satellite observations and ecosystem models reveal the presence of moderate blooms triggered by open ocean upwelling in the southwestern BoB (Vinayachandran and Mathew, 2003; Vinayachandran et al., 2005). In addition to the seasonal forcings, frequent occurrence of tropical cyclones favour short-lived isolated patches of intense blooms (Madhu et al., 2002; Vinayachandran and Mathew, 2003; Rao et al., 2006).

The biogeochemistry of the BoB has not been well explored and its biophysical interactions have received even lesser attention. Our present understanding of the mechanisms determining the spatial and temporal distribution of productivity in the BoB is limited, owing to the scarcity of observational data and model simulations. Ocean colour retrievals by satellites are widely affected by the presence of cloud cover during monsoon, the period when the bloom activity in the BoB is at its peak. Past observational studies (Vinayachandran and Mathew, 2003; Vinayachandran et al., 2004; Kumar et al., 2004; Kumar et al., 2007; Jyothibabu et al., 2015) have contributed to our understanding of the biological productivity in the BoB, suggesting that its bloom dynamics are complex, determined by the competing effects of winds (local as well as remote) and freshwater flux on the mixed layer processes. However, the spatial and temporal coverage of observations is insufficient to obtain a complete picture of the chlorophyll distribution. We also lack estimates of subsurface chlorophyll, and hence, its contribution to the column integrated productivity (Kumar et al., 2009) has received little attention.

Until now, the paucity of previous chlorophyll measurements precluded a detailed investigation of the bio-physical feedbacks and the possible controls on the surface properties, and air-sea heat and gas exchanges of the BoB. The present study is aimed at documenting the observed chlorophyll distribution of the southern bay, obtained from four ocean gliders and conductivity-temperature-depth (CTD) measurements, taken during the Bay of Bengal Boundary Layer Experiment (BoBBLE; Vinayachandran et al., 2018) field program. Surface bloom events in response to the monsoonal forcings at seasonal and synoptic timescales were observed at all glider locations. The BoBBLE data reveal the presence of prominent bloom activity at the subsurface of the BoB, which is rarely captured by satellites. Results from a coupled physical-ecosystem model are incorporated to evaluate the model performance in reproducing the summer blooms in the BoB and to analyse in detail the associated bio-physical interactions. Section 2 describes the observational data and the ecosystem model; Section 3 examines the vertical distribution of chlorophyll in the southern bay, colimited by light and nutrients, in response to the monsoonal wind and freshwater forcings. Summary and conclusions are given in the last section.

## 2 Observations and modelling

Observations were carried out in the region to the east coast of Sri Lanka, on-board ORV Sindhu Sadhana, which sailed from Chennai on 24 June 2016 and returned on 23 July 2016 (Fig. 1). The present analyses are based on the data along 8° N, extending from 85.3° E (hereafter referred to as TSW) to 89° E (hereafter referred to as TSE), including a 10–day CTD time series station at TSE. Shipboard measurements were taken back and forth along this longitudinal transect; the ship sailed from TSW to TSE during 29 June to





July, stayed at TSE from 03–15 July and returned back to TSW on 20 July. The longitudinal transect

runs across the productive regions of the SLD and SMC covering a distance of about 400 km.

## 2.1 In situ measurements of chlorophyll

Vertical distribution of chlorophyll was measured along the cruise track using ocean gliders and a ship-

board CTD. Ocean gliders are buoyancy driven autonomous underwater vehicles designed to dive from

the surface to the deep ocean and back following a sawtooth pattern, collecting vertical profiles of oceano-

graphic properties (Eriksen et al., 2001). Four gliders (SG579, SG534, SG532 and SG620) with biophysi-

cal sensors were deployed along the transect at $8°$ N (Fig. 1). They were positioned at specified locations,

hence the measurements made can be considered as time series data (SG579 shifted almost 60 km west-

wards during the observational period, but stayed within the SLD). The gliders provided high-resolution

measurements of biophysical properties, both in space (atleast 0.5 m in vertical) and time (4–7 profiles

a day). Data collection starts within the top 1m of the upper ocean, enabling better sampling of sur-

face properties compared with conventional measurement techniques. Each glider was equipped with a

SeaBird Electronics CTD package, a global positioning system (GPS), and Wetlabs Triplet ECOPuck

sensors. All ECOPucks had at least one fluorescence channel, measuring chlorophyll, and were accompa-

nied by one to two backscatter channels. In total, 405 dives were performed by the four gliders, including

shallow ($\sim$700 m) and deep ($\sim$1000 m) profiles, where each dive lasted 3–5 hours. The typical speed of

the gliders was about 0.25 m s$^{-1}$ and vertical velocities ranged between 0.10–0.15 m s$^{-1}$. The shipboard

CTD was equipped with auxiliary sensors for fluorescence, which are factory calibrated. In addition to

the gliders, the CTD collected a total number of 147 profiles along the cruise track. The CTD data used

for the present analysis is smoothed in time and depth spaces by 3 hours and 3 m respectively.

After quality control, the data from each glider were optimally interpolated (Bretherton et al., 1976)

onto a two-dimensional (time-depth) equally spaced grid, following Matthews et al. (2014). First, a back-

ground gridded field was constructed from a weighted average of the observations. A two-dimensional

Gaussian weighting function, with e-folding scales of 2 m for depth and 3 h for time, was used to map

each observation onto the depth-time grid. An optimal interpolation increment was then calculated, again

using the Gaussian weighting function, to calculate the final gridded field. The longitudinal positions

of the gliders were then used to create a single glider data set. The two dimensional (depth-time) op-

timally interpolated fields from each of the four gliders were combined into a single three-dimensional

(longitude-depth-time) gridded dataset, by linearly interpolating over longitude.

Observed fluorescence from gliders was corrected for non-photochemical quenching during daylight

hours using chlorophyll-to-backscatter ratios during night-time (Thomalla et al., 2018). The glider chloro-

phyll values exhibited an offset (Webber et al., 2014), with higher concentrations compared to the concur-

rent observations from the shipboard CTD. However, the glider data is reliable to explain the processes

underlying the bloom evolution since the spatial and temporal variability of chlorophyll were consistent

with the CTD observations. For the present analysis, the glider data corrected for non-photochemical

quenching was scaled to represent in situ chlorophyll value using the CTD data. An independent scale

factor was calculated for each glider's ECOPuck using linear regression with the available nearby CTD

profiles, where the distance between the ship and glider is not more than a quarter degree and the time

difference is not more than an hour.




## 2.2 Coupled physical-ecosystem model

A coupled physical-ecosystem model was employed to study the observed chlorophyll distribution in the southern BoB during the BoBBLE field program. The physical model is based on the Geophysical Fluid Dynamics Laboratory (GFDL) Modular Ocean Model Version 4 (MOM4p1, Griffies et al., 2004), configured for the Indian Ocean region extending from 30° E to 120° E and 30° N to 30° S (Kurian and Vinayachandran, 2007; Behara and Vinayachandran, 2016). Horizontal resolution of the model is 0.25° and the vertical grid spacing is 5 m in the upper 60 m, increasing to 10 m at 100 m depth, 20 m at 200 m depth, and 700 m at 5000 m depth, altogether forming 40 levels. The ETOPO5 dataset with 5 min resolution is used to set up the model topography, with the minimum depth of the ocean fixed at 30 m. A no-flux condition is applied across the model boundaries. Additionally, a no-slip condition is applied on the closed western and northern boundaries. The open southern and eastern boundaries consist of sponge layers where temperature and salinity fields are relaxed to climatology (Conkright et al., 1998) with a time scale of 30 days. The model mixing schemes are based on Large et al. (1994) and Chassignet and Garraffo (2001). Turbulent fluxes and upwelling longwave radiation are calculated using the bulk formula (Large and Yeager, 2004) and the penetrative shortwave radiation is parameterised based on Morel and Antoine (1994).

The ecosystem model used in this study is the Tracers of Phytoplankton with Allometric Zooplankton (TOPAZ) model (Dunne et al., 2010) consisting of 25 tracers including micro- and macro-nutrients, carbon, oxygen and lithogenic materials. The biogeochemical cycles are calculated with flexible nutrient stoichiometry. The phytoplankton class consists of three groups: small, large and diazotrophs. The small group represents the nanoplankton, which are weakly limited by nutrients and strongly limited by graz-

ing. The large group represents the microplankton, which are strongly limited by nutrients and weakly limited by grazing, with the ability to store iron internally. Diazotrophs (nitrogen fixers) form a relatively small fraction of the total biomass (Gnanadesikan et al., 2011). The model also includes dissolved organic matter and heterotrophic biomass. The biogeochemical mechanisms consist of nitrogen fixation, denitrification, gas exchange, atmospheric decomposition, scavenging and sediment processes. Co-limitation by light and nutrients controls the phytoplankton physiology and growth (Geider et al., 1997), with a temperature dependency (Eppley, 1972). Grazing is parameterized using a size-based relationship (Dunne et al., 2005), in which the large (small) phytoplankton group dominates the ecosystem at high (low) growth rates and biomass. Detritus production is temperature dependent and calculated as a fraction of phytoplankton (Dunne et al., 2005). Nitrification is inhibited by light (Ward et al., 1982). A detailed technical description of the ecosystem model is available in Dunne et al. (2010).

The model configuration used in the present analysis is similar to that in Thushara and Vinayachandran (2016). The physical model was spun up for a period of 10 years, starting from a state of rest using climatological initial fields for temperature and salinity (Conkright et al., 1998). This was followed by a coupled spin up for another 10 years, after switching on the ecosystem model. The succeeding interannual run was performed from 01 April 2015 to 31 December 2016. Nutrients for initialising the ecosystem model were obtained from the World Ocean Atlas (WOA09). The model forcing fields include air temperature, specific humidity, surface pressure, downward shortwave and longwave radiations, at hourly frequency from Goddard Earth Observing System (GEOS) Modern Era Retrospective-analysis for Research and Applications, Version 2 (MERRA-2 ; Rienecker et al., 2011). Wind speed and wind stress forcings were obtained from Advanced Scatterometer (ASCAT; Figa-Saldaña et al., 2002). The model freshwater

forcings include daily precipitation from Tropical Rainfall Measuring Mission (TRMM; Huffman et al., 2007) and monthly climatological river runoff from the Centre for Sustainability and the Global Environment (SAGE; Vörösmarty et al., 1996). Weekly chlorophyll from Sea-Viewing Wide Field-of-View Sensor (SeaWiFS; Sweeney et al., 2005) was used for the calculation of penetrative shortwave radiation.

## 3  Results and Discussion

The BoBBLE field program coincided with a suppressed phase of the Boreal Summer Intraseasonal Oscillation (BSISO), when the convective activity was weak over the southern BoB (see Figure 4 of Vinayachandran et al. (2018)). Precipitation was minimal during most of the observational period, until the establishment of the succeeding active phase of the BSISO by the end of the program. Surplus insolation associated with reduced atmospheric convection suggests that light availability only played a minor role in limiting the chlorophyll distribution, which makes the observational period ideal to study the bloom dynamics. Prior to the BoBBLE period, the region was characterised by increased cloud cover associated with the preceding active phase of the BSISO. Winds were stronger but solar insolation was lower, indicating significant light limitation on bloom generation during this period. Similar conditions re-established by the end of the BoBBLE period, in relation to the succeeding active phase. Monsoonal cloud cover, especially during the active phase of BSISO, limits the continuous sampling of ocean color from satellites, restricting the analysis of daily or weekly evolution of the blooms. Monthly means of chlorophyll obtained from European Space Agency (ESA) Ocean Colour Climate Change Initiative (OC-CCI v3.1) merged product reveal that the southern bay was biologically active during the BoBBLE period.





The mean chlorophyll concentration in the southern bay (82–92° E and 4–12° N) averaged for the month of July was about 0.2 mg m$^{-3}$, which is comparable to previous years.

## 3.1 Hydrography

To provide a dynamical context for the chlorophyll distribution, the hydrography of the southern BoB

during the BoBBLE period is briefly described here. Further details can be found in Vinayachandran et al. (2018) and Webber et al. (2018). In response to the prevailing atmospheric conditions, the upper ocean in the southern bay exhibited large spatial variability at seasonal and synoptic timescales. The climatological distribution of surface temperature shows cooler waters in the region of the SMC, creating an east-west contrast along 8° N (see Figure 1 of Vinayachandran et al. (2018)). Weaker winds and higher insolation,

associated with the suppressed phase of BSISO during the observational period, resulted in high sea surface temperature (SST). The mean SST obtained from the Group for High Resolution Sea Surface Temperature (GHRSST; Chao et al., 2009) dataset, averaged for the observational period (27 June–21 July 2016), was ∼29.3 °C at TSW and ∼0.5 °C less at TSE (not shown), deviating from the climatology. The mean sea surface salinity (SSS) from the Soil Moisture Active Passive (SMAP; Fore et al., 2016)

mission was ∼33.3 psu at TSW and farther east at TSE, salinity was 0.8 psu higher (not shown).

A depth-longitude section of temperature and salinity recorded by gliders, averaged for the period 03–14 July, is shown in Fig. 2. Gliders in the west (SG579 and SG532) exhibited higher SST and lower SSS compared to those in the east (SG534 and SG620), consistent with the satellite observations. The thermocline, represented by the 20 °C isotherm (D20), exhibits an east-west dip along 8° N extending

from TSW till 88° E, followed by a rise towards TSE (Fig. 2a). The western sector of the transect (TSW)

lies within the SLD, where open ocean Ekman pumping leads to the doming of the thermocline. At TSW, D20 is located at a depth of about 80 m, as observed by SG579, and deepens towards the east. In the region of the high salinity core of the SMC intrusion (Fig. 2b), D20 is much deeper, located at a depth of about 180 m (SG532). At the eastern end of the transect (TSE), D20 slightly shoals by about 40 m, as observed by SG620.

Circulation in the southern bay during the observational period is characterised by a strong cyclonic gyre in the region of the SLD and the monsoon current which flows north-eastward (Webber et al., 2018). During the beginning of the observational period, the SMC was strong with surface velocities ranging between 0.5-0.8 m s$^{-1}$ (Fig. 3a-g). The region of the SLD is characterised by strong negative sea level anomalies (SLA) of about -20 cm. By the end of the first week of July, the SMC weakened and shifted westward, reducing the zonal extent of the SLD (Fig. 3h-l). Farther east, towards the eastern edge of the monsoon current, the upper ocean was relatively less dynamic with weaker currents (0.1–0.3 m s$^{-1}$) and positive sea level anomalies (10–20 cm).

The spatial variability in the upper ocean dynamics of the BoB, determined by local and remote forcings associated with the monsoon, influence the biological response as well, which is of interest in the present study. The following sections characterise the observed chlorophyll in the southern bay in terms of intensities and the vertical distribution, during the BoBBLE period. The associated mechanisms determining the chlorophyll distribution are analysed, combining hydrographical observations and results from an ecosystem model.



### 3.2 Observed chlorophyll distribution

### 3.2.1 Surface bloom events

The gliders cover an east-west transect across the regions of the SLD and SMC (Fig. 1), providing time series measurements of chlorophyll. Surface layers remained weakly productive during most of the observational period, however, events of enhanced chlorophyll were observed at all the four glider locations (Fig. 4a-d) as well as in the CTD data (Fig. 4e). Surface chlorophyll concentrations from gliders and the CTD are shown in Fig. 5. During the beginning of the observational period, concurrent occurrence of surface blooms were observed within the SLD and along the path of SMC, as recorded by SG579, SG534 and SG532. At SG620 (TSE), two events were recorded with relatively weaker magnitudes. CTD measurements captured the surface blooms in the region of SMC during 01–02 July and at TSE during 06–08 July, consistent with the glider observations. The ship and glider were about 10 km apart during most of the observational period at TSE and hence an exact agreement in chlorophyll time series is not expected.

*Within the SLD*: Summer blooms in the region of the SLD have been reported earlier using satellite images of ocean color (Vinayachandran et al., 2004). The daily evolution of SLA and currents from Archiving, Validation, and Interpretation of Satellite Oceanographic data (AVISO) show the intensification of the SLD during the early phase (29 June–03 July) of the BoBBLE field program (Fig. 3). Observations from SG579, which falls right inside the dome, revealed the development of a surface bloom during the same period (Fig. 4a, 30 June–2 July). Chlorophyll concentration at the surface was $\sim 0.3$ mg m$^{-3}$ on 30 June, increased to $\sim 0.7$ mg m$^{-3}$ on 01 July and reduced to $\sim 0.4$ mg m$^{-3}$ on 02 July (Fig. 5a). CTD observa-



tions were available within the dome during 28–29 June, before the ship started moving eastwards from TSW. Until 29 June, surface chlorophyll values were much lower (< 0.1 mg m$^{-3}$), with higher concentrations mostly confined to a depth of about 30–60 m (Fig. 4e). Hence, it can be inferred that the surface blooms within the dome probably commenced on 30 June, peaked on 01 July and started decaying on 02 July. There were no glider observations of chlorophyll before 30 June to corroborate the CTD data.

The region of the SLD is characterised by negative SLA embedded within the cyclonic circulation to the east of Sri Lanka (Fig. 3). The hydrodynamics of the region suggests that the triggering mechanism for bloom generation is open ocean Ekman pumping forced by positive wind stress curl (Vinayachandran et al., 2004; Wijesekera et al., 2016a), favouring vertical transport of nutrients to the surface sunlit layers. The doming of the thermocline indicates dynamical uplifting of the nutricline and enhanced nutrient concentrations in the euphotic zone (Wilson and Coles, 2005; Turk et al., 2001). The thermocline was shallow, located at a depth of about 70 m, during the peak phase of the surface bloom (01 July, Fig. 4). The bloom event was characterised by lower surface temperatures (28.6 °C) and higher surface salinities (33.95 psu) with upsloping isotherms and isohalines (not shown), compared to the period when the surface chlorophyll concentrations were weak. The decay of surface bloom after 02 July (Fig. 5) followed the weakening of the dome (Fig. 3). Surface temperature increased by 0.7 °C and surface salinity decreased by ∼1.5 psu on 03 July, indicating the weakening of upwelling. CTD observations within the dome until 29 June, when the ship was at TSW, show that the subsurface chlorophyll concentrations were weak (< 0.5 mg m$^{-3}$) just before the surface bloom event (Fig. 4e). This indicates that the vertical redistribution of subsurface phytoplankton does not have significant contribution in enhancing the surface chlorophyll. The generation of surface blooms is presumed to be dominantly controlled by the vertical transport of subsurface nutrients



to the euphotic zone.

***Along the path of SMC***: Increased surface chlorophyll levels were observed at SG534 and SG532 during 1–2 July (Fig. 4b) and 2–4 July (Fig. 4c) respectively. Both gliders were located along the path of the SMC,

with SG532 in the region of the subsurface high salinity core (Fig. 2b). Surface chlorophyll concentration peaks at about 0.35 mg m$^{-3}$ and 0.4 mg m$^{-3}$ at SG534 and SG532 respectively (Fig. 5). The bloom events were associated with lower temperatures (28.7 °C and 29.1 °C for SG532 and SG534 respectively) and higher salinities (34.4 psu and 34 psu for SG532 and SG534 respectively) at the surface compared to the period when the surface blooms were absent.

Along the path of the SMC, the thermocline lies at deeper levels during the surface bloom events ($\sim$100–130 m at SG534 and $\sim$160–180 m at SG532), which is 40–100 m deeper than that in the region of dome (Fig. 4a-c). The spatial variability of thermocline is evident from the CTD observations as well, showing a shallow thermocline during the beginning (27–30 June) and end (20–21 July) of the field program, when the ship was in the west, and a deeper thermocline farther east (02–18 July; Fig. 4e). A deeper thermo-

cline generally indicates a deeper nitracline and stronger nutrient limitation in the surface layers. At the same time, the region of SMC is also subject to an additional supply of biologically rich waters advected from the coasts of India and Sri Lanka (Vinayachandran et al., 2004). In addition, the possibility of lateral advection of nutrients and chlorophyll generated within the SLD to the nearby glider locations cannot be ignored (see Section 3.3.2).

*Mixing events*: Chlorophyll distribution observed outside the dome, farther east at TSE, differed from that in the region of the SLD and SMC, in terms of intensity as well as the vertical structure. SG620 shows two events of surface blooms; the first event on 03 July and the second during 06–08 July (Fig. 4d). The surface chlorophyll concentrations were ∼0.3 mg m$^{-3}$ during the bloom events (Fig. 5). Both bloom events were characterised by cool surface temperatures and high surface salinities. The observed SST from SG620 was about 28.7 °C on 03 July and 28.8 °C on 06 July. Surface salinities were about 34.5 psu and 34.7 psu during 03 July and 06 July respectively. Temporal coverage of the first event is insufficient to explain its evolution since the bloom decays immediately after 03 July when the sampling begins. Wind speed measured by the shipboard automatic weather station (AWS) was 5–9 m s$^{-1}$ during 03 July (Fig. 6). A deeper mixed layer depth (MLD) of about 60 m during the bloom event indicates that vertical mixing is the primary factor which favoured the increase in surface chlorophyll. The second event was captured by the CTD measurements as well (Fig. 4e), consistent with the glider data. This event coincided with a phase of increasing wind speed of about 6–11 m s$^{-1}$ (06–07 July; Fig. 6). Subsequent deepening of the mixed layer (∼70 m, Fig. 4d) suggests the role of mixing and entrainment in triggering the surface blooms. Enhanced vertical processes favour intensification of surface chlorophyll by transporting nutrients to the euphotic zone and by redistributing the subsurface chlorophyll to the surface layers.

Intermittent occurrence of freshening events were observed at the surface, associated with local precipitation and lateral advection, the latter being prominent. The decay period of the bloom (08–10 July) coincided with the development of a freshening event. Surface salinity decreased by about 0.8 psu from 06 July to 10 July (Fig. 6) and the corresponding decrease in surface chlorophyll was about 0.27 mg m$^{-3}$



(Fig. 5). There was an overall reduction in total chlorophyll integrated over the mixed layer by about 20 mg m$^{-2}$ (Fig. 6). Freshening could be attributed to the lateral advection of low saline waters from the nearby regions, since no local rainfall was observed during this period. Vertical profiles of temperature, salinity and chlorophyll from SG620 during different stages of the surface bloom evolution are shown in

Fig. 7a-c. During the peak of the surface chlorophyll bloom (06 July), the mixed layer was deep ($\sim$55 m), with an almost uniform distribution of bio-physical properties (Fig. 7a), and the isothermal layer depth coincided with the MLD. The following days (08–10 July) were characterised by strong salinity stratification with the arrival of freshwater in the surface layers. The mixed layer shoaled to $\sim$30 m (Fig. 7b and c), whereas the isothermal layer remained around the same depth. The associated development of a barrier

layer is noticeable, with a thickness of $\sim$25–30 m. CTD observations at TSE also captured this freshening event and the subsequent decay of the surface bloom (Fig. 7d-f). With the arrival of freshwater, surface salinity as recorded by the CTD decreased by about 0.5 psu and the mixed layer shoaled by about 25 m, creating a strong barrier layer (Fig. 7f). The corresponding decrease in surface chlorophyll was $\sim$0.15 mg m$^{-3}$.

Freshening and the barrier layer formation inhibit the development of phytoplankton blooms in the surface layers by restricting vertical transport of subsurface nutrients and chlorophyll. Even though high wind speed ($\sim$10–12 m s$^{-1}$) conditions prevailed during the decay period of the bloom, freshwater induced stratification was strong enough to overcome the wind effect (Fig. 6). The observed biological response to freshwater is similar to that in the northern bay, where salinity stratification restrains the growth of

phytoplankton by inducing nutrient limitation in the surface layers (Kumar et al., 2002).



### 3.2.2 Deep chlorophyll maxima

Chlorophyll maxima at the subsurface are indicative of active biological productivity beneath the surface layers of the ocean. The formation of deep chlorophyll maxima (DCM) is determined by a variety of mechanisms including enhanced growth rate of phytoplankton colimited by light and nutrients at optimum depths, photoacclimation of pigment content, and physiologically controlled swimming behaviors and buoyancy regulation (Cullen, 2015). The BoB is reported to have prominent DCM (Murty et al., 2000; Madhu et al., 2006), which contribute to the column integrated productivity (Gomes et al., 2000; Madhupratap et al., 2003; Li et al., 2012) with magnitudes often comparable to the highly productive Arabian Sea (Kumar et al., 2009). However, little is known about the distribution of subsurface chlorophyll in the BoB and the associated processes, due to the lack of observations.

During the BoBBLE field program, both the glider and CTD observations revealed the presence of prominent DCM in the southern bay (Fig. 4 and Fig. 8a). The chlorophyll maxima were centered at a depth of about 20–50 m, mostly below the mixed layer and above the thermocline (Anderson, 1969). Similar depth ranges of DCM were reported previously by Gomes et al. (2000) and Kumar et al. (2009) in the BoB. Subsurface chlorophyll concentrations range from 0.3–1.2 mg m$^{-3}$ (Fig. 8a), which were 2-3 times higher than the surface values (Fig. 5). DCM were prominent in the region of the SLD and along the path of the SMC (Fig. 4a-c), whereas outside the dome, the subsurface concentrations were weaker (Fig. 4d).

Vertical profiles of chlorophyll from the gliders during events of enhanced surface chlorophyll are shown in Fig. 9. The mean DCM was intense, located at a depth of about 20–30 m, in the region of the SLD and the SMC (Fig. 9a-c). The DCM became weaker, diffused and slightly deeper (30-40 m) at TSE (Fig. 9d and


e). Intensification of DCM in the region of SLD can be related to the doming of thermocline, followed by an upward sloping of nutricline. A shallow nutricline enriches the euphotic zone with limiting nutrients, enhancing the growth of phytoplankton. At TSE (SG620) thermocline was deeper, indicating a deeper nutricline and stronger nutrient limitation in the euphotic zone. During the surface bloom events, mixing

often penetrated to deeper layers pushing the mixed layer towards the DCM (Fig. 4d and e). This favours the dilution of DCM and a decrease in phytoplankton concentration at the subsurface through mixing with the weakly productive surface layers, leaving a near homogeneous distribution of chlorophyll within the water column (Fig. 9d and e).

Subsurface chlorophyll concentrations were noticeably higher in the region of the SMC (Fig. 4c and

Fig. 9c). Maximum intensities were recorded by SG532, with magnitudes ranging from 0.7–1.2 mg m$^{-3}$ during 02–07 July (Fig. 8a). Column-integrated chlorophyll was also observed to be the highest at SG532 (04 July), with total chlorophyll in the top 100 m reaching as high as 35 mg m$^{-2}$ (Fig. 8b), which is comparable to the previously observed values in the BoB (Gomes et al., 2000; Madhupratap et al., 2003; Kumar et al., 2009; Li et al., 2012). The region of SMC is characterised by the advection of upwelled

chlorophyll rich water from the west coast of India and the southern coast of Sri Lanka. An isolated maximum (1.2 mg m$^{-3}$) in the DCM was recorded by SG579 in the region of SLD in the later half of the observational period (15 July). However, in the absence of surface blooms, the corresponding column-integrated chlorophyll was lower (28 mg m$^{-2}$), compared to the region of the SMC.

The core subsurface intrusion of the SMC, below the low salinity surface waters of the southern bay

was located around SG532 during the observational period (Vinayachandran et al., 2018; Webber et al., 2018). The vertical salinity structure reveals a high salinity core at 88° E, extending up to a depth of about



180 m, with salinity values as high as 35.8 psu (Fig. 2b). Arabian Sea water, which is rich in nutrients and chlorophyll sliding through the subsurface layers of the BoB, is presumed to be contributing to the intensification of the DCM at SG532, suggesting a key role of SMC intrusion in the biological budget of the southern bay. However, it may be noted that the location of the subsurface high salinity core was

much deeper relative to the depth of DCM. Most of the high salinity intrusions at 88° E occured below 80 m, in the deeper layers of the euphotic zone. Dynamics behind the distribution of the DCM in the region of the high salinity core are intricate. Though the effect of lateral advection by the SMC on DCM cannot be ignored, the possible contribution of vertical processes in supplying the subsurface nutrients or chlorophyll needs to be examined in detail.

Subsurface chlorophyll concentrations were observed to intensify for shorter durations following the weakening of surface blooms (Fig. 4). Increases in DCM concentrations after the decay of surface blooms were about 0.13 mg m$^{-3}$, 0.37 mg m$^{-3}$ and 0.25 mg m$^{-3}$ at SG534, SG532 and SG620 respectively (Fig. 8a). In the region of the SLD (SG579), the subsurface chlorophyll concentrations increased to ~0.7 mg m$^{-3}$ during the peak phase of the surface bloom (01 July). During the decaying phase of the surface

bloom (02-05 July), these high chlorophyll levels (0.7 mg m$^{-3}$) were maintained at the subsurface and weakened afterwards (Fig. 8a). This indicates enhanced biological productivity at the subsurface, after the triggering mechanisms inducing the surface blooms have weakened. During the decaying phase of surface blooms, the upper layers of the water column became less turbulent or more stably stratified (Fig. 7), inhibiting the vertical transport of nutrients and chlorophyll. However, the subsurface layers

still possess enough nutrients to support phytoplankton growth. For example, the surface bloom event at SG620 weakened in response to the freshening event on 08 July (Fig. 7b). Consequently, there was an





increase in DCM, which lasted for a period of about 2-3 days from 10–12 July (Fig. 8). The observed intensification of DCM in the absence of surface chlorophyll can be explained in terms of changes in subsurface irradiance levels. During the decaying phase of the surface bloom, the self-shading effect of surface phytoplankton weakens, enhancing the light availability at the subsurface, which is examined in

the following section.

### 3.2.3 Role of light limitation

Chlorophyll interactive penetrative radiation was calculated at TSE for the period 04-14 July, following Morel and Antoine (1994) and Manizza et al. (2005) scheme as given below,

$$I_{(z)} = I_{IR} \cdot e^{-k_{IR}z} + I_{RED(z-1)} \cdot e^{-k_{(RED)}\Delta z} + I_{BLUE(z-1)} \cdot e^{-k_{(BLUE)}\Delta z}$$

$I_{(z)}$ is the penetrative radiation at each depth level, $I_{IR} = I_0 \cdot (0.58)$ represents the infrared band, $I_{VIS} = I_0 \cdot (0.42)$ represents the visible band and $k_{IR} = 2.86$ m$^{-1}$ is the light attenuation coefficient for the infrared band. The self-shading effect of phytoplankton is taken into account so that at every vertical level (z), the available visible light is computed as a function of irradiance at the level just above (z-1). $\Delta z$ is the thickness of each layer between two vertical levels, which is 1 m in the present glider data. Visible light

is splitted in two averaged wavelength bands as given below,

$$I_{RED} = I_{BLUE} = \frac{I_{VIS}}{2},$$

where $I_{RED}$ and $I_{BLUE}$ are the irradiances in red and blue/green bands respectively.

The light attenuation coefficients for the two visible bands is calculated as a function of chlorophyll concentration ([Chl]) as follows,

$$k_{(RED)} = 0.225 + 0.037 \cdot [Chl]^{0.629}$$



$$k_{(BLUE)} = 0.0232 + 0.074 \cdot [Chl]^{0.674}$$

Surface irradiance ($I_0$) for the above calculations was obtained from shipboard AWS (Fig. 10a) and chlorophyll from SG620 (Fig. 4d). In order to exclude the effect of daily variation in surface irradiance, a diurnal composite of radiation (Fig. 10a) for the period 4–14 July is also used for the calculations.

Estimated penetrative radiation and the depth of the euphotic zone using observed surface irradiance and the diurnal composite are shown in Fig. 10b and c respectively. The depth of the euphotic zone is estimated as the depth where irradiance reduces to 1 W m$^{-2}$ (Pastor et al., 2013). Nearly 40–60 % of the radiation was absorbed in the top 1 m of the water column and 80–90 % in the top 30 m. Below the DCM, irradiance levels were substantially weaker (< 10 W m$^{-2}$). During the daylight hours of peak insolation, the euphotic zone extended to 70–110 m, with a well defined diurnal cycle.

The depth of euphotic zone was least (70–80 m) during the surface bloom event (06–07 July), indicating enhanced absorption of radiation in the surface layers (Fig. 10b). Euphotic depth calculated using the diurnal composite of irradiance also shows a minimum during the same period (Fig. 10c). The shoaling of the euphotic zone during the bloom event indicates the self-shading effect of surface phytoplankton. Enhanced attenuation of radiation by near-surface phytoplankton reduces the irradiance levels in the deeper layers and strengthens the light limitation on phytoplankton growth in the subsurface. As a result, bloom activity weakens in the subsurface layers, despite the availability of nutrients.

Following the decay of surface blooms owing to nutrient limitation, the euphotic zone depth increased due to the penetration of radiation to deeper layers (Perry et al., 2008). The deepening of the euphotic zone following the decay of the surface bloom was about 25 m on 08 July (Fig. 10b). Enhanced light availability in the subsurface layers favours the intensification of DCM (Fig. 4d,e and Fig. 7). It should be noted that

the DCM may not represent a deep biomass maximum as photoacclimation (Cullen, 1982; Geider, 1987;

Mateus et al., 2012) leads to changes in carbon to chlorophyll ratios. At the base of the euphotic layer, the

cellular concentration of chlorophyll will increase as an adaptation to the lower irradiance levels (Cullen,

2015).

## 3.3 Model simulation

A coupled physical-ecosystem model, employed to study the aforementioned bloom features in the BoB-

BLE region, enabled further understanding of the three-dimensional mixed layer processes controlling the

evolution of chlorophyll blooms. The role of horizontal advection by the SMC and dynamics of the SLD

in determining the simulated distribution of nutrients and chlorophyll is analysed in detail. The model

provides a fairly good representation of the bio-physical features in the BoB. The physical model repro-

duces the observed seasonal and intraseasonal features of the Indian Ocean, with a realistic representa-

tion of the mixed layer processes and the heat and freshwater budgets (Kurian and Vinayachandran, 2006;

Kurian and Vinayachandran, 2007; Vinayachandran and Kurian, 2007; Behara and Vinayachandran, 2016).

Basin-averaged SST in the BoB (80–100° E and 0–25° N) for the month of July is about 28.37 °C, with

a cold bias of 0.85 °C compared to the GHRSST observations. The seasonal temperature distribution of

the southern bay, including the cooling associated with upwelling off the coasts of India and Sri Lanka

and the development of the cold pool, is well represented. The model reproduces the low salinity plumes

associated with freshwater influx in the northern bay and high salinity intrusions from the Arabian Sea

into the southern bay. Mean surface salinity for the basin is about 32.59 psu for the month of July, which

exceeds SMAP observations by about 0.6 psu. The intrusion of the SMC into the BoB and its bifurcation





into several branches is reproduced by the model. The vertical distribution of salinity reveals intermittent occurrence of high salinity cores at deeper levels, associated with the subsurface intrusion of the SMC. The model reproduces a well-developed SLD, characterised by negative SLA (-10 cm) embedded within the cyclonic circulation east of Sri Lanka, consistent with the AVISO observations.

The TOPAZ ecosystem model simulates well the mean distribution of oceanic productivity (Sarmiento et al., 2010; Pastor et al., 2013; Marvasti et al., 2016) and the biophysical interactions associated with major climatic events including Indian Ocean Dipole, El Niño Southern Oscillation and Atlantic Multidecadal Oscillation (Park and Kug, 2014; Park et al., 2014; Gnanadesikan et al., 2014). The model provides a realistic representation of the monsoonal biophysical interactions in the Indian Ocean and has been used to ex-

plain the bloom dynamics of northwestern BoB during the summer monsoon (Thushara and Vinayachandran, 2016) and northeastern Arabian Sea during winter (Vijith et al., 2016).

For the present analysis, simulated surface chlorophyll is validated using monthly means obtained from the OC-CCI merged product. The observed spatial distribution of surface chlorophyll, averaged for the month of July, to be consistent with the BoBBLE period, is shown in Fig. 11a. Along the path of the SMC,

a distinct band of moderate blooms is present with concentrations of about 0.3–0.6 mg m$^{-3}$. The band extends from the southern coast of Sri Lanka up to about 11° N and 89° E, indicating lateral transport of nutrients and chlorophyll carried by the SMC from the upwelling regions off the coasts of India and Sri Lanka. Seasonal evolution of chlorophyll in the region of the SLD is not well captured by the satellites, probably because of gaps in the ocean colour retrieval during the peak phase of the dome (29 June to

02 July). Moderate blooms (0.2–0.3 mg m$^{-3}$) are observed in regions farther east and southeast of the monsoon current.

### 3.3.1 Simulated chlorophyll distribution

The observed spatial distribution of surface chlorophyll in the BoB is well represented by the model
(Fig. 11b), with prominent blooms in the coastal ocean, northwestern bay and the southern bay (Vinayachandran,
2009). Bloom intensities are highest along the coastal regions, with magnitudes exceeding 1 mg m$^{-3}$. The

northwestern bay is characterised by the seasonal occurrence of upwelling blooms triggered by coastal
Ekman pumping and advection towards the offshore regions (Thushara and Vinayachandran, 2016). The
southern bay exhibits an isolated patch of chlorophyll in the region of the SLD and moderate blooms along
the path of the SMC. Surface bloom concentrations are about 0.6–0.7 mg m$^{-3}$ and 0.3–0.4 mg m$^{-3}$ in
the region of the SLD and the SMC respectively. The model blooms are generally weaker compared with

satellite observations. The bias can be attributed either to the deficiencies in external nutrient inputs in the
model or the overestimation of coastal blooms by satellites in the presence of optically active constituents
other than chlorophyll (Gregg and Casey, 2004; Blondeau-Patissier et al., 2014). The presence of DCM
is well represented by the model, consistent with the glider and CTD observations. Realistic representa-
tion of the chlorophyll distribution indicates that the model is good at simulating monsoonal bio-physical

interactions in the BoB.

While the major seasonal features of the southern BoB are reproduced by the model, they are often not
exactly at the observed locations. For example, the SLD is slightly shifted westward and the meandering
of the SMC around Sri Lanka is weaker (Fig. 11c and d), probably due to the discrepancies in the model
wind forcing or the simulated remote forcings. The eastward (northward) extension of surface chlorophyll

associated with the SMC is overestimated (underestimated). These inaccuracies can be ignored while
examining the large-scale seasonal features, but may be significant at mesoscales or smaller scales. Hence,

the ecosystem model results are used to explain the biological response to seasonal features including the Sri Lanka dome and the monsoon current, in comparison with the concurrent observations from gliders (SG579, SG534 and SG532) and the shipboard CTD.

The model SLD develops around 85° E, 8° N, close to the sampling location of SG579. A longitudinal transect extending from 82° E to 92° E along 8° N is selected to examine the vertical distribution of temperature, salinity, nitrate and chlorophyll on 01 July, during the peak phase of the surface chlorophyll bloom in the region of the SLD (Fig. 12). The region is characterised by an intense bloom ($\sim$ 0.5–0.8 mg m$^{-3}$) at the surface and a prominent DCM ($\sim$ 0.5-1.2 mg m$^{-3}$) centered at a depth of about 20-30 m (Fig. 12a), well below the mixed layer (Fig. 12b). Temperature profiles show upsloping isotherms, providing cooler (27 °C) waters to the surface layers (Fig. 12b). Similarly, the salinity distribution shows increased surface salinity (33.5 psu) with isohalines shoaling to the surface (Fig. 12c). Doming of the thermocline (D20) is evident between 83–87° E along the transect (Fig. 12b). The thermocline rises to a depth of $\sim$60 m, which is about 80 m shallower than the nearby regions outside the dome.

The dynamics of the SLD favour biological productivity through the vertical transport of nutrients triggered by open ocean upwelling (Vinayachandran et al., 2004; Vinayachandran, 2009). The modelled bloom intensifies during the peak phase of the dome and decays with the weakening of the dome, consistent with the BoBBLE observations. The nitracline shoals (Fig. 12a) along with the vertical displacement of isotherms and isohalines. We prefer using the 2 $\mu$mol kg$^{-1}$ nitrate isoline as the nitracline rather than the vertical gradient criterion, since the absolute concentration of nutrients available for phytoplankton uptake is more important for bloom generation than the gradients (Wilson and Coles, 2005). The euphotic zone is enriched with high nitrate concentrations in excess of 10 $\mu$mol kg$^{-1}$. The DCM shoals to about 30



m, which is 10–30 m shallower than the nearby regions. By the second week of July, cyclonic circulation in the region of dome weakens and shifts towards the northwest. The subsequent reduction in Ekman pumping leads to the decay of the bloom due to nutrient limitation.

Chlorophyll distribution in the region of the SMC is influenced by the horizontal advection of both nutrients and chlorophyll. Simulated surface nitrate shows enhanced concentrations along the path of the SMC, indicating the lateral advection of nutrient-rich waters from the Arabian Sea (Fig. 11d). Advection of chlorophyll from the upwelling regions off the coasts of India and Sri Lanka could further intensify the bloom concentration (Vinayachandran et al., 2004; Vinayachandran, 2009). The relative role of mixed layer processes in maintaining the summer blooms along the path of SMC is presented in Section 3.3.2.

The model DCM shows large spatial variability in terms of intensity and depth. The DCM is strong in the region of the SLD and along the path of the SMC (Fig. 13a), consistent with the glider observations (Fig. 4). Subsurface chlorophyll concentrations increase to about 1.2 $\mu$mol kg$^{-1}$ within the dome, which is more than twice the concentrations outside the dome. At the same time, the depth of the DCM is minimum in the region of the SLD (Fig. 13c). The DCM shoals to $\sim$20 m within the dome and deepens to $\sim$70 m outside the dome. Productivity is closely correlated with SLA and the depth of the nitracline and thermocline (Signorini et al., 1999; Wilson and Coles, 2005; Sarma, 2006; Signorini et al., 2015). The strongest DCM (Fig. 13a) coincides with the shallowest nitracline (Fig. 13d). Ekman pumping leads to the upsloping of nitracline, which increases the concentration of limiting nutrients in the euphotic zone. The column integrated chlorophyll is found to be maximum along the path of the SMC (Fig. 13b), with magnitudes ranging from 50-70 $\mu$mol kg$^{-1}$.



### 3.3.2 Mixed layer nutrient budget

The nutrient budget from the ecosystem model is examined to identify the relative roles of mixed layer processes in controlling the summer blooms in the southern BoB. In the TOPAZ ecosystem model, the growth of phytoplankton is determined by a limiting nutrient, in a multinutrient environment. Here, inorganic nitrate ($NO_3$) concentration is used to represent the nutrient budget (Fig. 14), since the dominant role of nitrate in controlling the biological productivity of the BoB is well known (Kone et al., 2009). The observed nitrate distribution has been used in previous studies to explain phytoplankton distribution in the BoB (Kumar et al., 2002, 2004, 2007). The present simulation also shows that during the pre-monsoon period, productivity in the southern bay is largely limited by nitrate when mixed layer dynamics were less favourable for the vertical supply of nutrients to the surface sunlit layers. Hence $NO_3$ was preferred over $PO_4$ and Fe ($SiO_4$ does not limit growth in TOPAZ) to explain the nutrient distribution. In addition, the chlorophyll concentration in TOPAZ is proportional to the nitrogen in phytoplankton (Dunne et al., 2010). Total chlorophyll is calculated as,

$$Chl = C\!:\!N \cdot 12 \cdot 10^6 \cdot (\theta^{Sm} \cdot N^{Sm} + \theta^{Lg} \cdot N^{Lg} + \theta^{Di} \cdot N^{Di}),$$

where $C\!:\!N$ is the carbon to nitrogen ratio, $12 \cdot 10^6$ is the molecular mass of carbon in $\mu g\ mol^{-1}$, $\theta$ is the chlorophyll to carbon ratio ($Chl\!:\!C$), and N is the phytoplankton nitrogen concentration in $mol\ kg^{-1}$.

Physical processes controlling the model nutrient distribution include horizontal advection and vertical processes (including vertical advection and mixing). The biological processes include a source term represented by nitrification and sink terms comprising denitrification and uptake by the phytoplankton. The time rate of change of nitrate is given by,

$$\frac{\partial NO_3}{\partial t} = -\nabla \cdot \mathbf{u} NO_3 + \nabla K \nabla NO_3 + S_{NO_3},$$



where **u** is the velocity vector from the OGCM, K is the vertical diffusivity and $S_{NO_3}$ represents the biological processes.

Weekly averages of the model nitrate budget terms averaged over the mixed layer from 24 June to 21 July, comprising the BoBBLE observational period are shown in Fig. 14. The model MLD is defined as the depth at which the buoyancy difference with respect to the surface is equal to 0.0003 m s$^{-2}$. Before the onset of the summer monsoon, the upper ocean of the southern BoB maintained oligotrophic conditions, where nutrient levels were weak, inhibiting the growth of phytoplankton (not shown). Mixed layer dynamics associated with the monsoonal forcings play a dominant role in controlling the nutrient distribution of the southern BoB. As the monsoon intensifies, the monsoon current becomes stronger and the cyclonic circulation off the east coast of Sri Lanka leads to the development of the Sri Lanka Dome (Fig. 11).

The last week of June, coinciding with the beginning phase of the BoBBLE observational period, was characterised by a developing phase of the SLD, with strong open ocean upwelling. Nitrate concentrations in the mixed layer increased (Fig. 14a) as a result of enhanced vertical transport (Fig. 14b). At the same time, these nutrients were transported away from the region of upwelling and redistributed to the nearby regions through horizontal advection (Fig. 14c). Along the southern tip of India and Sri Lanka, coastal upwelling driven by alongshore winds leads to the intensification of nitrate levels, as evident from the vertical processes (Fig. 14b). Offshore transport of upwelled nutrients occurs at significant rates, enhancing the nitrate concentrations in regions away from the coast (Fig. 14c). Within the mixed layer, uptake by the phytoplankton is higher than nitrification, so that the sink term exceeds the source term. Hence, biological processes contribute to a reduction in total nitrate, mainly in the coastal ocean and the region of SLD, where phytoplankton concentrations are high (Fig. 14d).

During the first week of July, nitrate levels in the mixed layer reduced slightly compared to the previous week; this period was characterised by the gradual weakening of the SLD and a reduction in the vertical supply of nutrients (Fig. 14f), leading to a decline in nitrate levels. Consequently, the associated horizontal transport (Fig. 14g) to the nearby regions also reduced. The nitrate uptake reduced due to the reduction in phytoplankton concentration, which explains the weaker negative tendencies due to biological processes in the region of the dome (Fig. 14h). Upwelling along the coasts of India and Sri Lanka (Fig. 14f) and the offshore advection effects (Fig. 14g) were still prominent during this period.

During the second week of July, nitrate levels in the mixed layer were generally higher compared to the previous week, especially in the region of the SMC (Fig. 14i). The SLD slightly regained its strength till 10 July, and weakened immediately. The related vertical transport of nutrients intensified (Fig. 14j) and the upwelled nutrients were distributed to the nearby regions (Fig. 14k). Though the upwelling was not as strong as that in the preceding peak phase (during the last week of June), vertical supply of nitrate occurred at higher rates (Fig. 14b and j). As a result of strong upwelling in the preceeding peak phase of the SLD, the nitrate isolines became shallower (not shown). This preconditioning probably favoured enhanced vertical supply of nitrate to the surface layers during the second peak phase, though the strength of upwelling was weaker.

The simulated eastward velocities associated with the summer monsoon current off the southern coast of India and Sri Lanka strengthened during the second week of July in relation to increasing wind speeds. Along the path of SMC, a clear patch of increased nitrate levels was evident (Fig. 14i), which extended from the southern tip of India up to about 85° E. This indicates horizontal advection of coastally upwelled nutrients from the southern coasts of India and Sri Lanka (Fig. 14k) into the southern BoB by the SMC.

Lateral supply of nutrients by the SMC supports the growth of phytoplankton along its path. Increased uptake of nitrate by the phytoplankton further enhanced the negative contribution of biological processes (Fig. 14l).

During the third week of July, nitrate levels along the path of the SMC decreased (Fig. 14m). Following a reduction in wind speed, the monsoon current off the southern coast of India weakened and so did the horizontal transport (Fig. 14o). Vertical supply of nutrients was maintained in the region of dome (Fig. 14n). Contribution by biological processes decreased as the nitrate uptake weakened following a reduction in phytoplankton concentration (Fig. 14p). In summary, the above analyses show that the distribution of nutrients and the biological productivity in the southern BoB is largely dependent on the mixed layer dynamics associated with the summer monsoon and the relative roles of vertical and horizontal processes vary spatially following the circulation features.

## 4  Summary and Conclusions

The BoB plays a major role in controlling the monsoon variability through its unique upper ocean properties (Gadgil et al., 1984; Vecchi and Harrison, 2002; Shankar et al., 2007). A deeper understanding of the bio-physical feedbacks in the BoB is of primary importance since oceanic productivity plays a major role in modifying the air-sea heat and gas exchanges (Arrigo et al., 1999; Chisholm, 2000). Despite its climatic significance, estimates of chlorophyll distribution in the BoB are limited owing to the restrictions in spatio-temporal coverage of in situ data sampling. Remote sensing of ocean color is widely affected by the monsoonal cloud cover and turbid nature of coastal waters. In the presence of salinity stratification, which imparts strong nutrient limitation in the surface layers, intense bloom activity is mostly confined to

the subsurface layers of the BoB. Hence, satellite retrieval algorithms based on ocean color in the surface

layers would lead to an underestimation of actual chlorophyll content in the water column. These limi-

tations in data sampling imply the need for high resolution and sustained measurements of the vertical

distribution of chlorophyll in the BoB.

In this paper, we document the observed vertical distribution of chlorophyll in the southern BoB during

the BoBBLE field program conducted during the summer monsoon of 2016. High-resolution data sam-

pling using gliders accompanied by shipboard CTD record prominent bloom activity in the southern BoB,

with persistent DCM at intermediate depths. Hydrographic features of the region suggest that the observed

spatio-temporal distribution of chlorophyll is strongly linked to the competing effects of monsoonal wind

and freshwater forcings, which control the light and nutrient limited growth rate of the phytoplankton. Re-

duced atmospheric convection and surplus insolation during the observational period suggest that surface

chlorophyll distribution is weakly limited by light and dominantly determined by the nutrient availabil-

ity. On the other hand, subsurface chlorophyll distribution is controlled by the balance between light and

nutrient limitations.

The present observations underline the previously reported (Vinayachandran et al., 2004; Vinayachandran,

2009; Jyothibabu et al., 2015) role of the SLD and the SMC as the major physical drivers determining the

biological productivity of the southern BoB. The region of the SLD is characterised by enhanced chloro-

phyll concentrations in the presence of a shallow thermocline (nitracline). A distinct band of chlorophyll

is observed all along the path of the SMC, highlighting the role of lateral advection of nutrient-rich waters

from the Arabian Sea in enriching the oligotrophic upper ocean of the BoB. In addition to the seasonal

forcings, intermittent mixing events induced by local wind forcing trigger surface chlorophyll blooms

outside the dome. A coupled physical-ecosystem model simulates satisfactorily the aforementioned distribution of chlorophyll, with prominent blooms in the regions of SLD and SMC. Model nutrient budget analyses demonstrate the role of monsoon dynamics in controlling the spatial and temporal distribution of biological productivity in the southern BoB. Open ocean Ekman pumping of nutrients is identified as

the major factor trigerring the generation and maintenance of summer blooms in the region of the SLD. On the other hand, advection by the SMC supplies coastally upwelled nutrients along the southern coasts of India and Sri Lanka to the southern BoB, favouring enhanced bloom concentrations.

Prominent bloom activity is observed at the subsurface indicating the contribution of DCM in the column-integrated productivity of the BoB, where the surface waters are generally oligotrophic. Intense

DCM exist in the region of the SLD and the SMC, whereas outside the dome, subsurface blooms are weaker. Spatial variability of DCM intensity indicates that the dynamic uplifting of the thermocline (nutricline) is more efficient in enriching the euphotic zone with nutrients compared with wind-induced mixing. Upwelling leads to a sharp and intense DCM, whereas mixing results in a more diffuse and weaker DCM. The region of the subsurface intrusion of the SMC exhibits the strongest DCM among all the glider

locations, suggesting the contribution of Arabian Sea water in the biological budget of the BoB.

Inhibition of surface blooms induced by the freshwater effect was often observed in the southern BoB during the study period, similar to that in the northern BoB. The intermittent occurrence of surface freshening events favour restratification of the upper ocean and formation of barrier layers. Stratification curtails the wind-induced vertical transport of nutrients and subsurface chlorophyll, leading to the decay of

surface blooms. Meanwhile, freshening leads to an intensification of DCM, favoured by enhanced light penetration into deeper layers as the self-shading effect weakens in the absence of surface chlorophyll

blooms. In addition, shoaling of the mixed layer induced by salinity stratification impedes the vertical

redistribution of subsurface chlorophyll, thereby intensifying the DCM.

The shape of chlorophyll profiles in different dynamical regimes indicates that the processes deter-

mining the vertical distribution of chlorophyll are intricate, which needs to be explored in detail using

comprehensive datasets. The intensity and depth distribution of DCM depends on a wide range of factors

including the hydrography of the upper ocean, biochemical nutrient cycling as well as the physiological

adaptations of different phytoplankton communities. The deep chlorophyll maxima do not necessarily

represent biomass maxima, since the chlorophyll-to-biomass ratio varies with different phytoplankton

species as well as with nutrient and light availability at depths (Geider et al., 1997, 1998; Wang et al.,

2009; Li et al., 2010). Other loss terms including grazing and mortality rates also have to be taken into

account for a complete description of the evolution of chlorophyll blooms.

Bio-physical interactions in the ocean have significant impacts on climate variability through the control

on upper ocean dynamics (Morel, 1988; Sathyendranath et al., 1991; Murtugudde et al., 2002; Strutton and Chavez,

2004; Manizza et al., 2005). Understanding different aspects of oceanic productivity helps to determine

the potential feedbacks on the climate system. Proper estimation of the vertical distribution of marine

phytoplankton and the total chlorophyll content in the upper ocean will help to understand the strength

of carbon cycling in the ocean. Apart from the climatic impacts, the global marine fisheries production is

highly dependent on the seasonal distribution of phytoplanton in the major fishing zones. Advanced data

sampling using gliders, designed to operate under adverse oceanic conditions can make significant contri-

butions in the understanding of biogeochemical cycling of the ocean and its climatic impacts, implying the

need for expanding such observations for future research. Realistic simulation of monsoonal bio-physical

interactions underlines the potential role of ecosystem models in exploring the vertical distribution of oceanic productivity, which is beyond the scope of satellites.

*Data availability.* The data sets analysed in this paper are available from the corresponding author upon reasonable request.

*Author contributions.* V. Thushara and P. N. Vinayachandran performed data analysis and manuscript preparation. Bastien Y. Queste, Ben-
jamin G. M. Webber and Adrian J. Matthews performed the glider data correction and quality control. V. Thushara carried out the ecosystem model simulation. All the authors contributed significantly in data interpretation.

*Competing interests.* No competing interests are present.

*Acknowledgements.* BoBBLE is a joint MoES, India–NERC, UK program (MM/NERC-MOES-02/2014/002). The BoBBLE field experi-
ment on board RV Sindhu Sadhana was funded by Ministry of Earth Sciences, Govt. of India under the Monsoon Mission program adminis-
tered by Indian Institute of Tropical Meteorology, Pune. We are thankful to the captain, technicians and crew of RV Sindhu Sadhana for their support and co-operation. BGMW was supported by the NERC BoBBLE project (NE/L013827/1). Glider data were processed using the UEA glider toolbox (http://www.byqueste.com/toolbox.html). We thank Global Modeling and Assimilation Office (GMAO) and the Goddard Earth Sciences Data and Information Services Center (GES DISC) for providing MERRA reanalysis product (http://disc.sci.gsfc.nasa.gov/daac-bin/DataHoldings.pl), IFREMER (http://www.ifremer.fr/cersat/en/data/data.htm) for ASCAT winds, AVISO (http://www.aviso.altimetry.fr)
for SLA, ESA OC-CCI v3.1 (http://www.esa-oceancolour-cci.org/) for chlorophyll, JPL PODAAC (https://podaac.jpl.nasa.gov) for SST and SSS products, TRMM (http://daac.gsfc.nasa.gov/precipitation) for precipitation and SAGE (http://www.sage.wisc.edu/riverdata/) for river

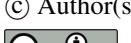



discharge. Thanks to GFDL for providing the source code of the coupled physical-ecosystem model (MOM4P1-TOPAZ). Computations

were carried out at the Super Computer Education and Research Centre, Indian Institute of Science, Bangalore. Ferret has been used for data

analysis and graphics.



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





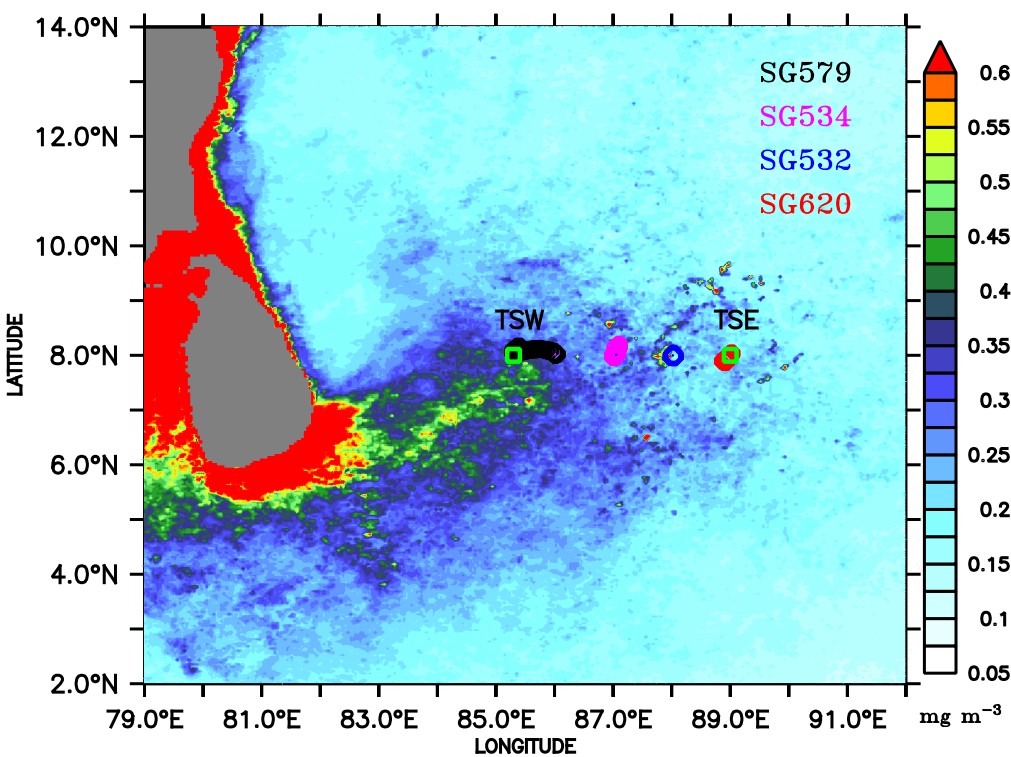

**Figure 1.** Chlorophyll (mg m$^{-3}$) climatology (2007–2016) for the month of July obtained from Ocean Colour Climate Change Initiative (OC–CCI) version 3.1. Ocean glider locations are marked as circles along 8° N, where the shipboard observations were performed. The glider deployment locations are (8° N, 86° E), (8° N, 87° E), (8° N, 88° E), and (8° N, 88°54′ E) for SG579, SG534, SG532, and SG620 respectively. Observational period of gliders are 30 June–20 July, 01–17 July, 02–16 July, and 03–14 July of 2016 for SG579, SG534, SG532, and SG620 respectively. TSW and TSE (squares) are sampling locations at (8° N, 85.3° E) and (8° N, 89° E) respectively.




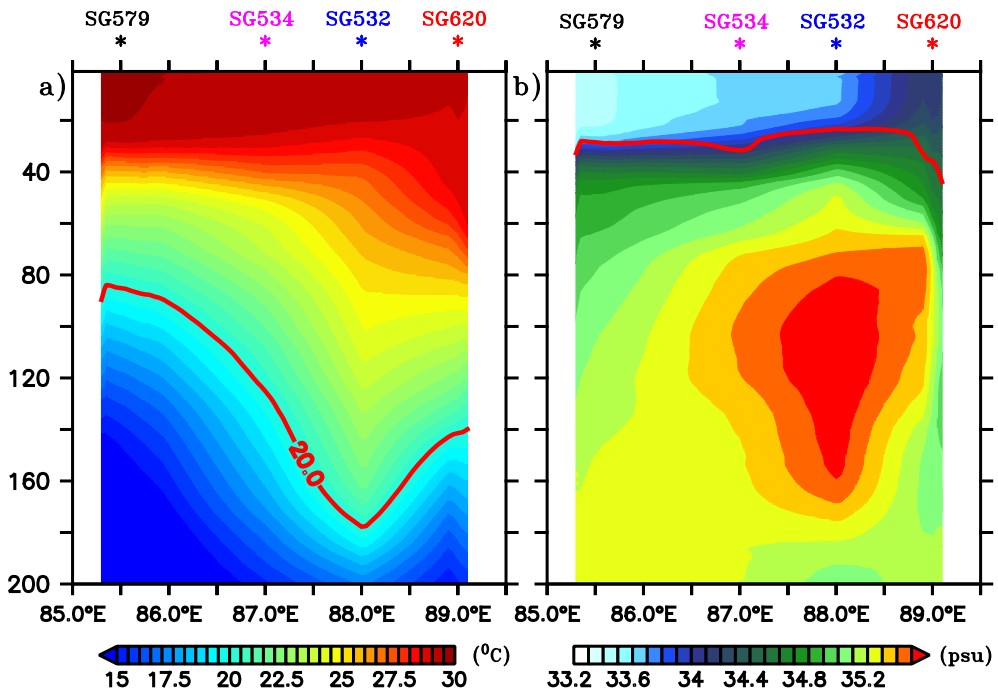

**Figure 2.** Depth-longitude sections of a) temperature (°C) and b) salinity (psu) obtained from ocean gliders averaged for 03–14 July, the

common period when all the gliders performed data sampling. Mean glider locations are marked at the top of each panel. Red curves in a)

and b) represent the thermocline and MLD respectively. The thermocline is represented by the 20 °C isotherm (D20). MLD is calculated as

the depth where density is equal to the sea surface density plus an increase in density equivalent to 0.8 °C.





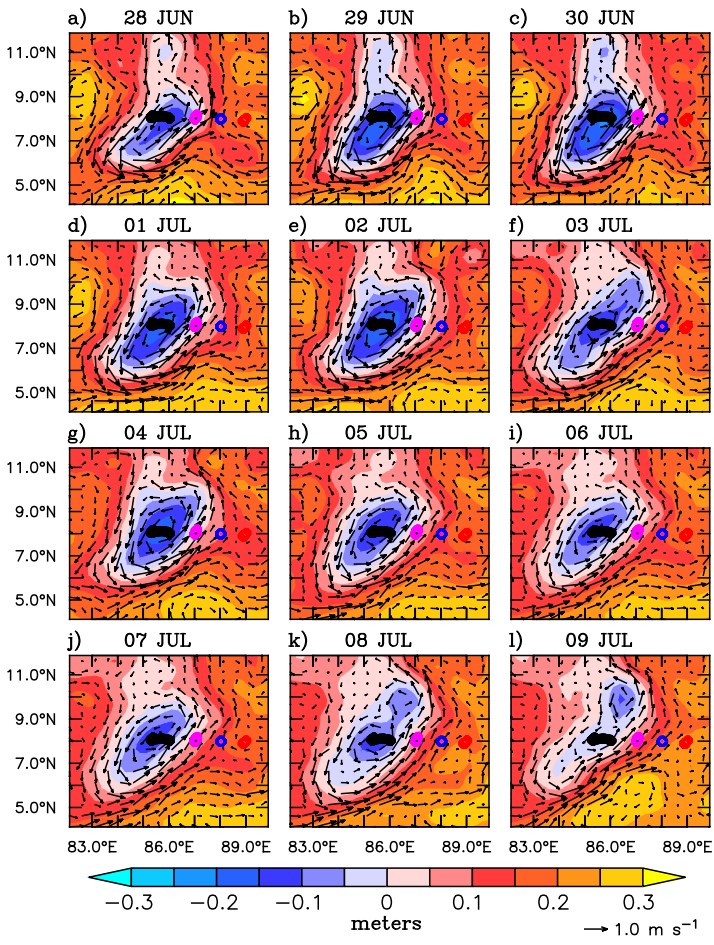

**Figure 3.** Sea level anomalies (SLA; m) and surface currents (m s$^{-1}$) from AVISO for the period 28 June 2016 to 09 July 2016. The glider locations are marked along 8° N (circles). Evolution of Sri Lanka dome (SLD) is represented by the negative SLA embedded within the cyclonic circulation.





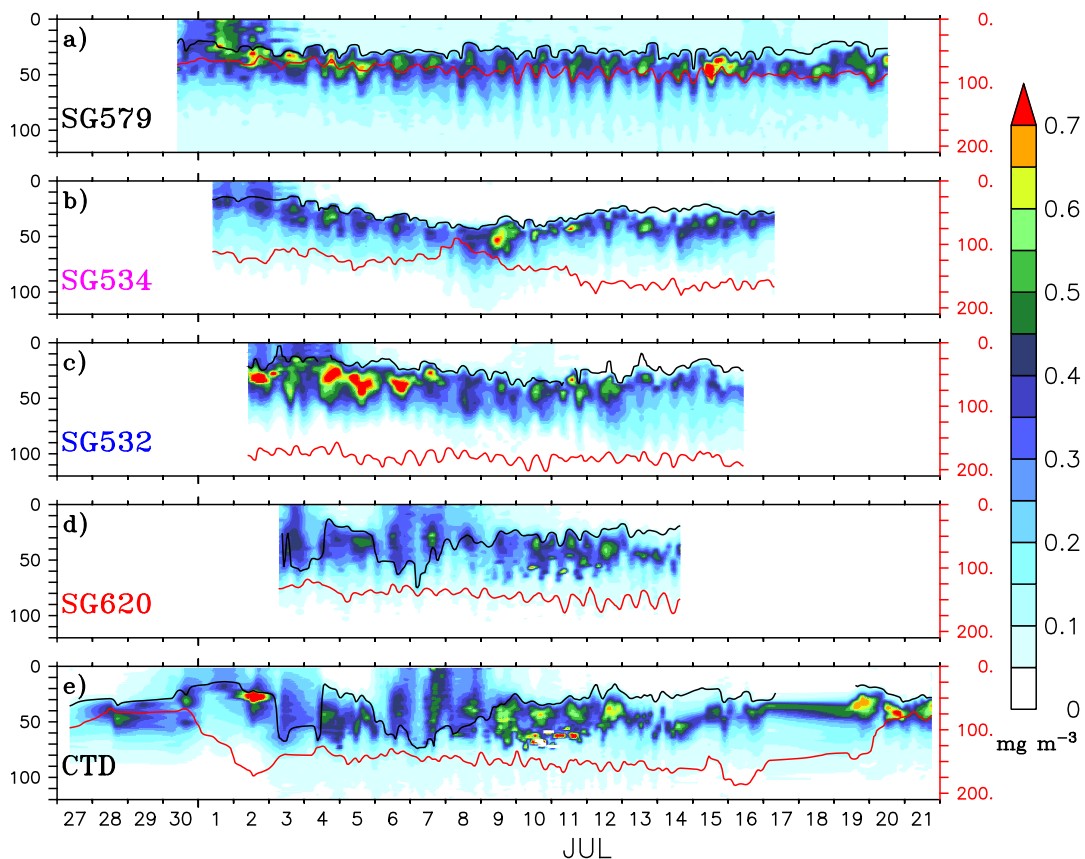

**Figure 4.** Time-depth sections of chlorophyll (mg m$^{-3}$) from ocean gliders (a-d) and CTD (e). The glider measurements are considered as

time series data for the locations shown in Figure 1. CTD observations were collected at TSW (85.3° E, 8° N) from 27 June to 29 June, after

which the ship sailed towards TSE (89° E, 8° N). From 03–15 July, time series measurements were made at TSE, after which the ship sailed

back towards the west and reached TSW on 20 July. The black curve represents the mixed layer depth, which is calculated as the depth where

density is equal to the sea surface density plus an increase in density equivalent to 0.8 °C. The thermocline (red curve) is represented by the

20 °C isotherm (D20). Note that the y-axis at the right side has a different scale.



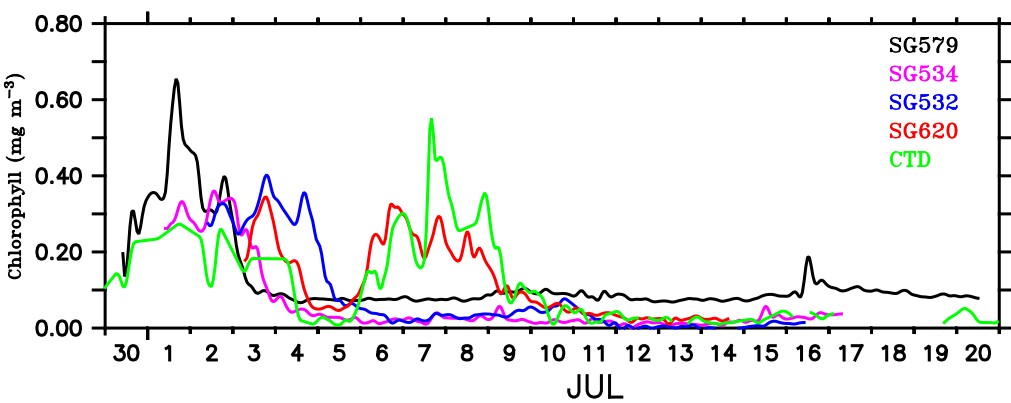

**Figure 5.** a) Surface chlorophyll concentration (mg m$^{-3}$) from ocean gliders (at 1 m) and the shipboard CTD (at 3 m). SG579 (black) falls

within the region of SLD, SG534 (magenta) and SG532 (blue) along the path of SMC and SG620 (red) at the outer edge of SMC as shown

in Figure 1. CTD (green) observations were collected along the 8° N section as described in Figure 4.





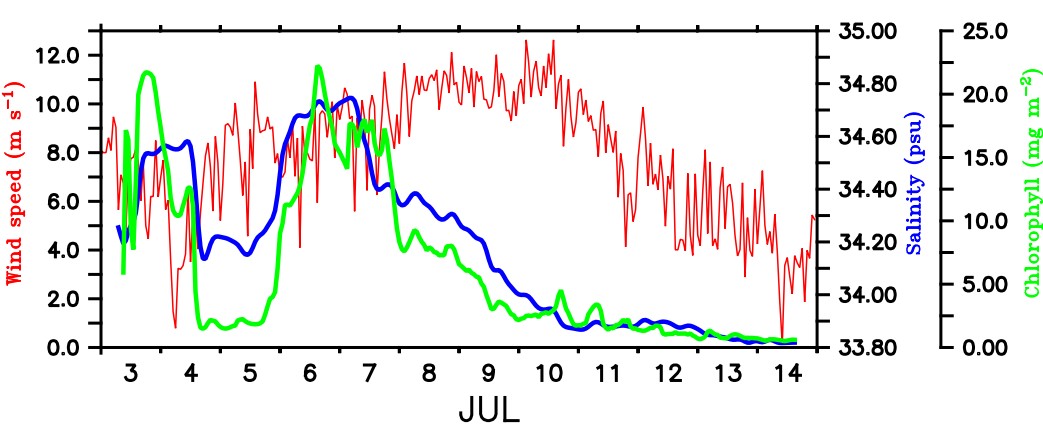

**Figure 6.** Time series of wind speed (m s$^{-1}$; red) from shipboard AWS at TSE (89° E, 8° N). Surface salinity (psu; blue) and total chlorophyll

integrated over the mixed layer (mg m$^{-2}$; green) is from SG620 deployed at TSE. MLD is calculated as the depth where density is equal to

the sea surface density plus an increase in density equivalent to 0.8 °C.





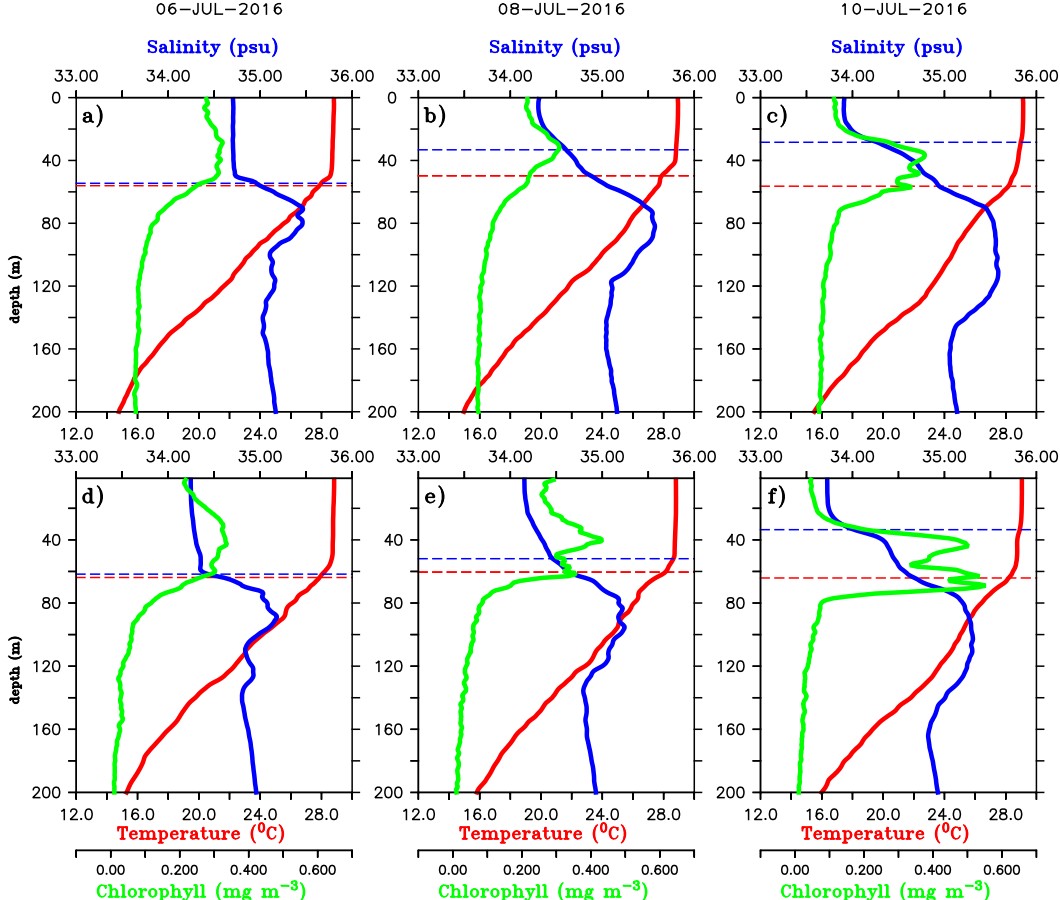

**Figure 7.** Daily mean vertical profiles of temperature (°C; red), salinity (psu; blue) and chlorophyll (mg m$^{-3}$; green) during 06 July (left panels), 08 July (middle panels) and 10 July 2016 (right panels) from SG620 (top panels) and CTD (bottom panels). The blue dashed line indicates the mixed layer depth, which is calculated as the depth where density is equal to the sea surface density plus an increase in density equivalent to 0.8 °C. The red dashed line indicates isothermal layer depth (ILD) which is calculated as the depth where the temperature is cooler than SST by 0.8 °C. The region between the MLD and ILD represents the barrier layer.





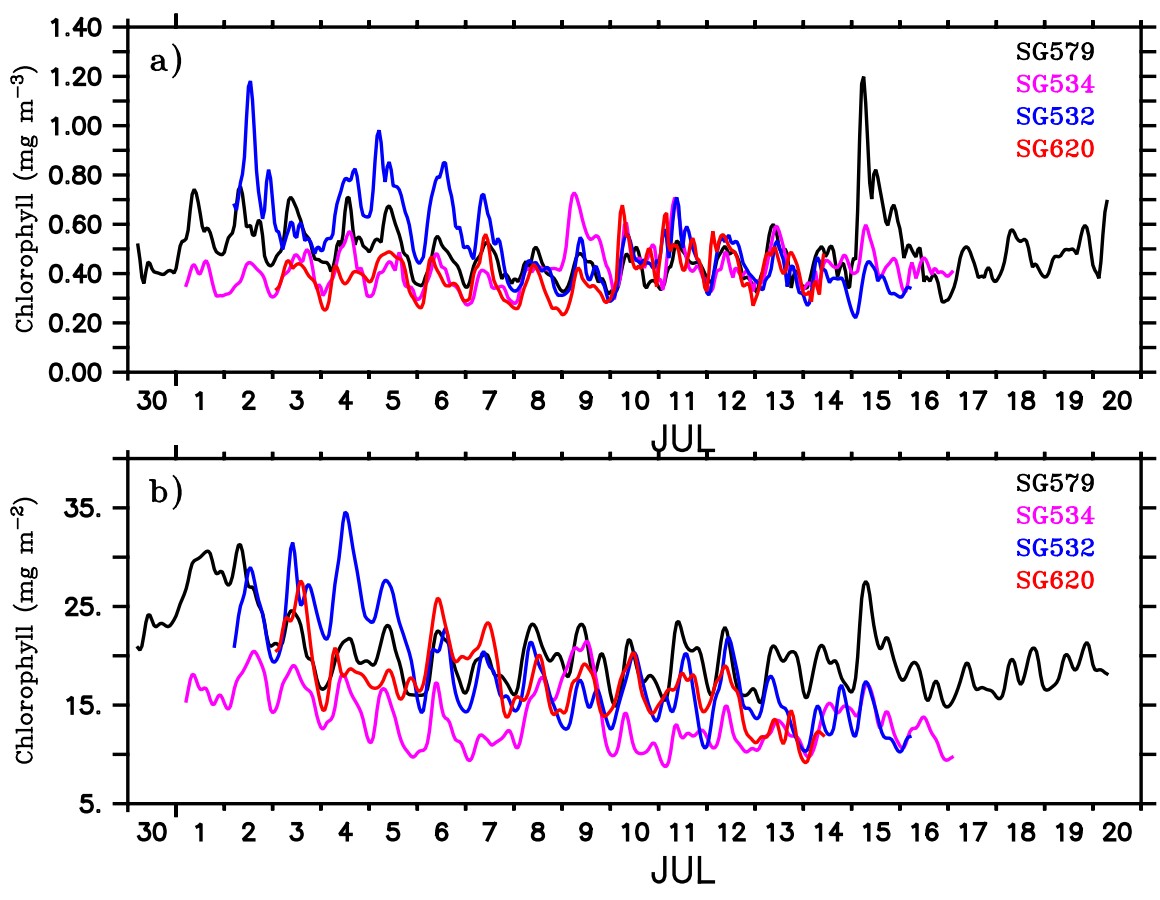

**Figure 8.** a) Concentration of deep chlorophyll maxima (mg m$^{-3}$) and b) depth-integrated (100 m) chlorophyll (mg m$^{-2}$) from ocean gliders;

SG579 (black), SG534 (magenta), SG532 (blue) and SG620 (red).





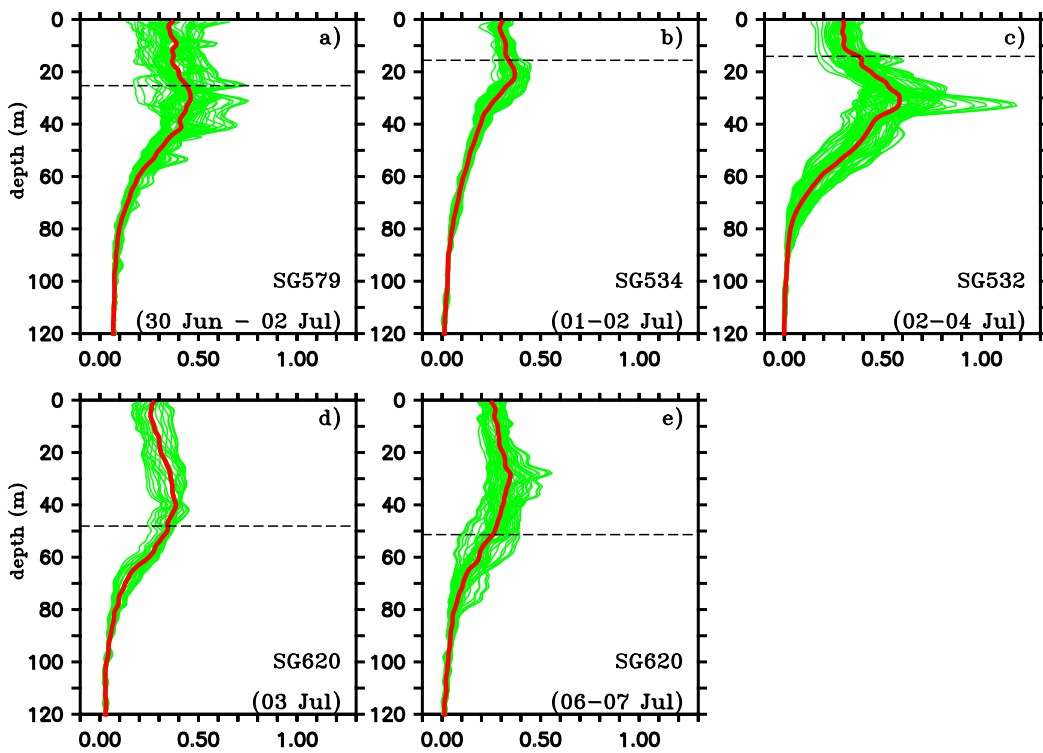

**Figure 9.** Vertical profiles of chlorophyll (mg m$^{-3}$) from ocean gliders during surface bloom events as shown in Fig. 5. Individual profiles are given in green and the corresponding mean profiles in red. Black dashed line represents the mixed layer depth, which is calculated as the depth where density is equal to the sea surface density plus an increase in density equivalent to 0.8 °C.



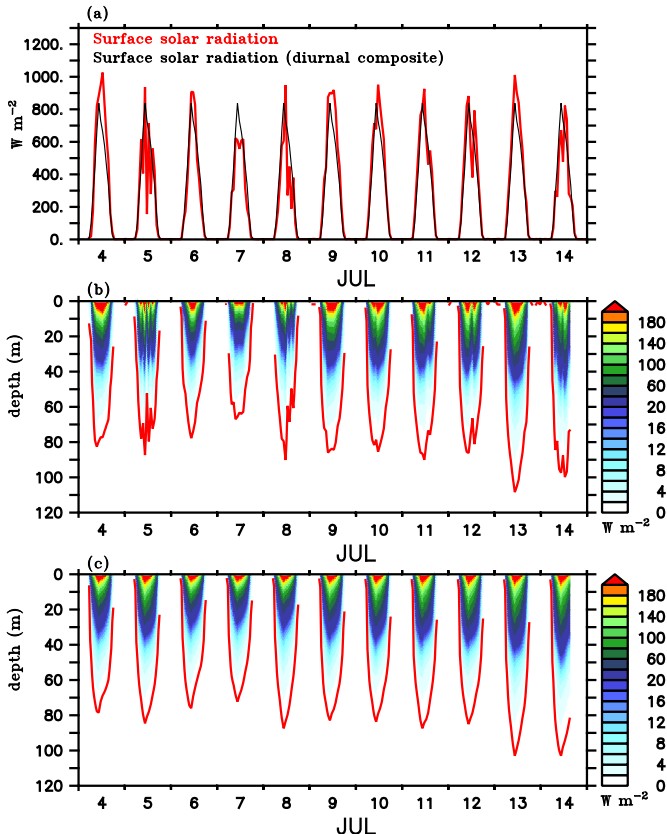

**Figure 10.** (a) Surface solar radiation measured by the shipboard AWS at TSE from 04–14 July (red) and the corresponding diurnal composite (black) calculated for the same period. Penetrative shortwave radiation (W m$^{-2}$) calculated following Morel and Antoine (1994) and Manizza et al. (2005) scheme using (b) observed and (c) diurnal composite of radiation. Chlorophyll from SG620 is used for the calculations. The red curves in b) and c) represents the depth of euphotic zone (m) which is taken as the depth of 1 W m$^{-2}$ irradiance.





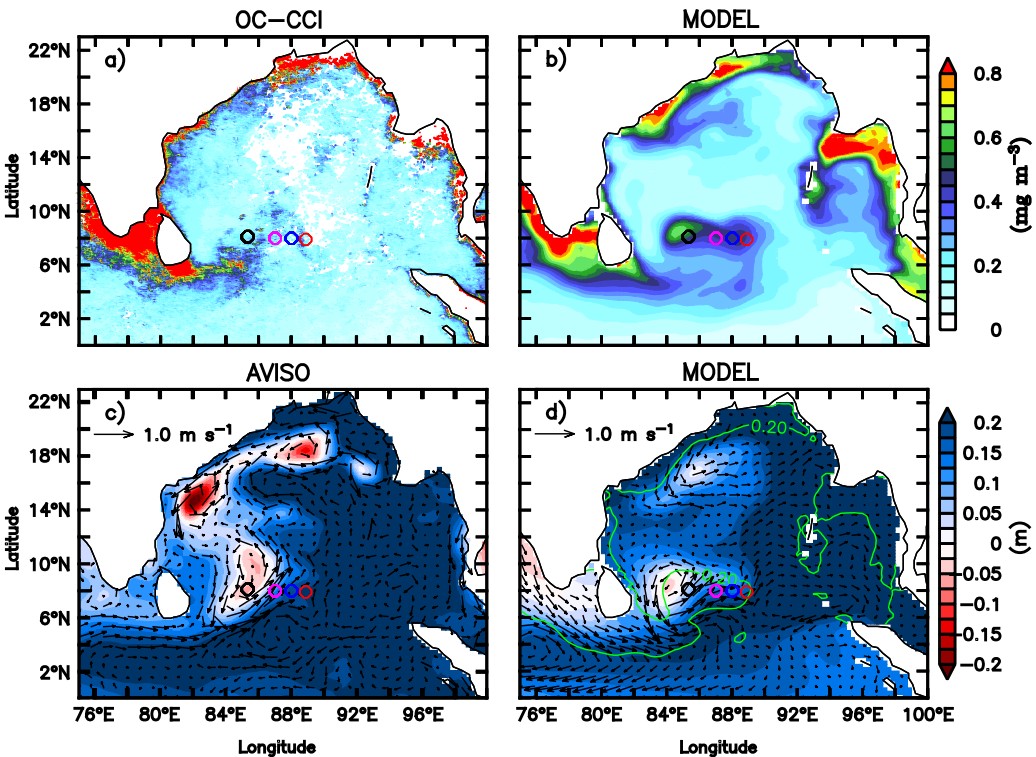

**Figure 11.** Comparision of the coupled physical-ecosystem model simulation with observations. Monthly mean surface chlorophyll concentrations (mg m$^{-3}$) for July 2016 from a) ESA OC-CCI merged product and b) model. Monthly mean SLA (m) are overlayed with surface current (m s$^{-1}$) vectors from c) AVISO and d) model. Green contour in panel d) represents 0.2 mmol kg$^{-1}$ nitrate isolines. The glider locations are marked as circles in the study region along 8° N.





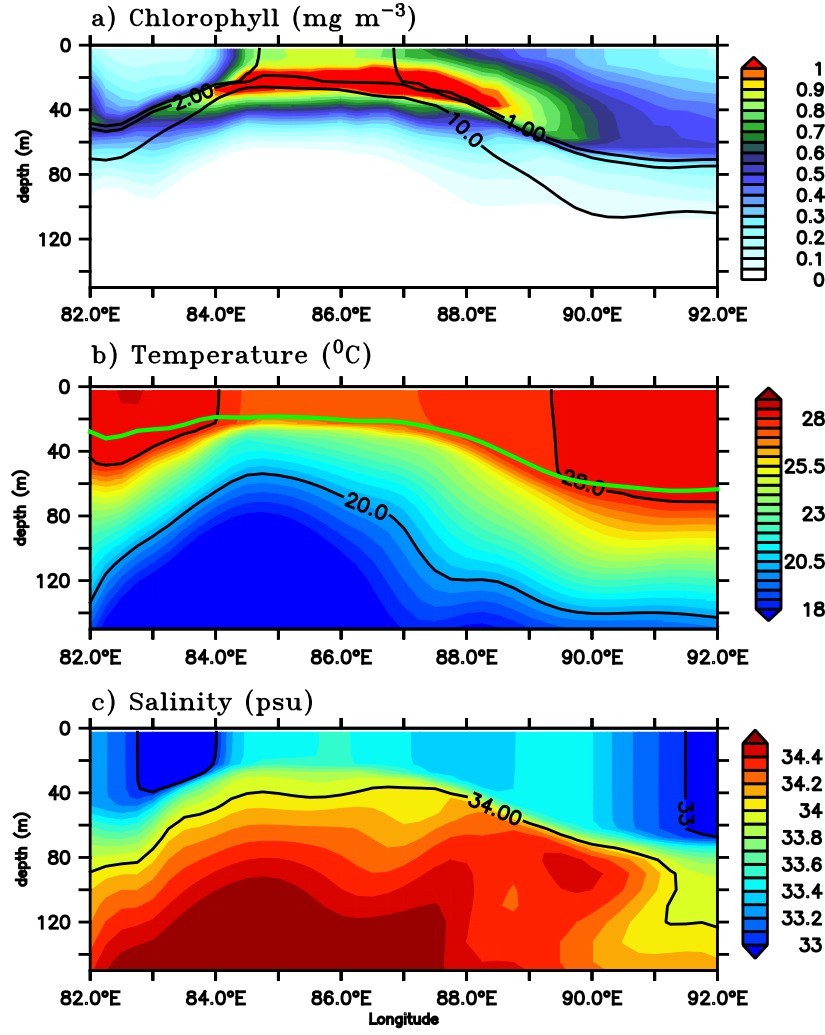

**Figure 12.** Depth-longitude sections of a) chlorophyll (mg m$^{-3}$), b) temperature (°C) and c) salinity (psu) along 8° N for 01 July 2016 from the ecosystem model. Black contours in panels a), b) and c) represent nitrate (1 $\mu$mol kg$^{-1}$, 2 $\mu$mol kg$^{-1}$ and 10 $\mu$mol kg$^{-1}$), temperature (20 °C and 28 °C) and salinty (33 psu and 34 psu) respectively. Green curve in panel b) represents the model mixed layer depth.





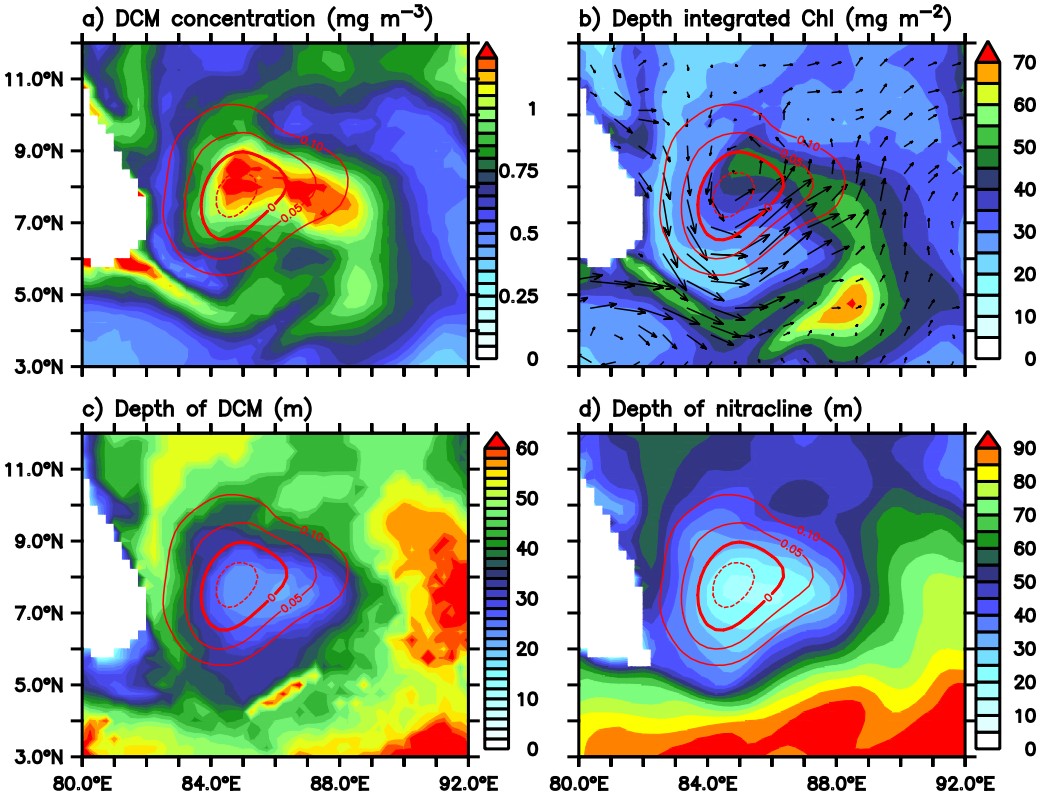

**Figure 13.** a) Intensity of deep chlorophyll maxima (DCM; mg m$^{-3}$), b) depth-integrated (100 m) chlorophyll (mg m$^{-2}$), c) depth of DCM (m), and d) the depth of nitracline (m) for 01 July 2016 from the ecosystem model. Nitracline is defined as the depth of 2 $\mu$mol kg$^{-1}$ nitrate isoline. Red contours in all the panels represent SLA (m) in the region of the Sri Lanka dome.




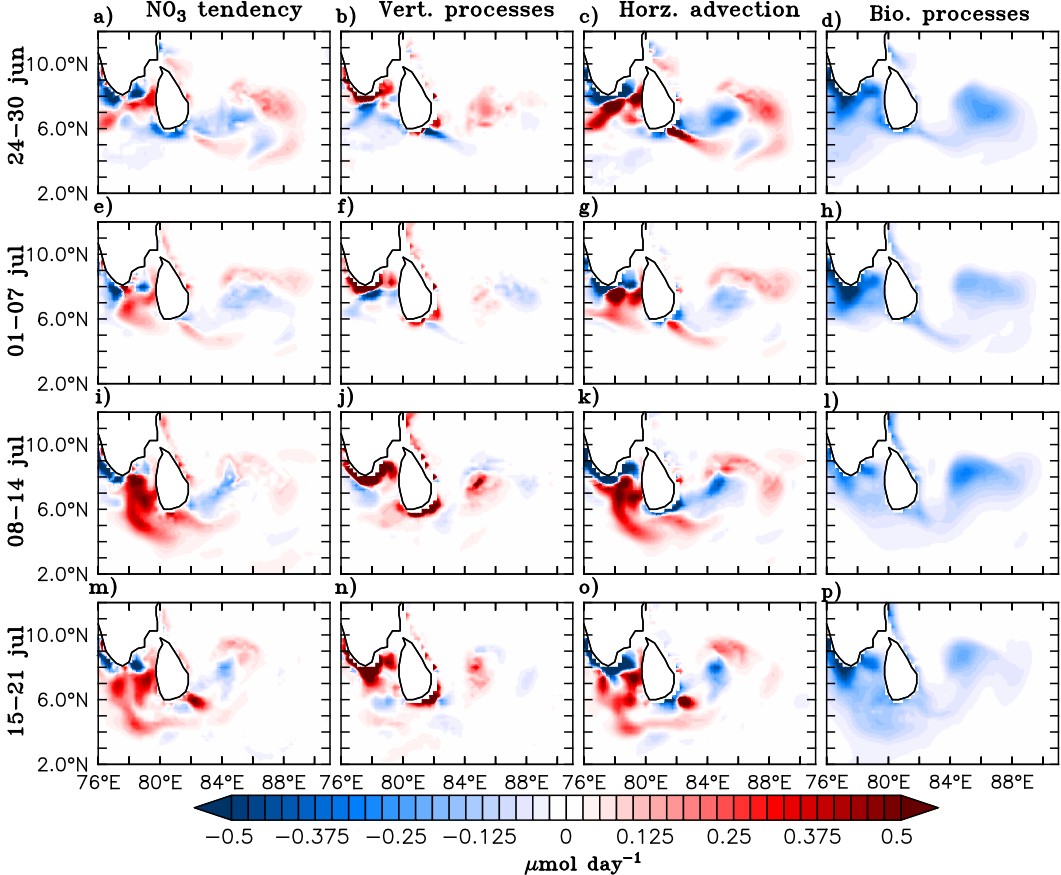

**Figure 14.** Model nitrate budget averaged over the mixed layer. Nitrate tendency (first column), vertical processes (second column), horizontal advection (third column) and the biological processes (fourth column) in $\mu$mol day$^{-1}$ are shown for 7-day averages starting from 24 June to 21 July 2016, marked on the left side of the corresponding panels. Vertical processes include vertical advection and mixing, and biological processes include source (nitrification) and sink (denitrification and uptake by the phytoplankton) terms for the model nitrate.