# Peer review of "Vertical distribution of chlorophyll in dynamically distinct regions of the southern Bay of Bengal"

_Biogeosciences, 2018_

## Referee Comment (RC1) · Anonymous Referee #1 · 13 Aug 2018

Review of "Vertical distribution of chlorophyll in dynamically distinct regions of the southern Bay of Bengal" by Thushara et al.

Based on an observational campaign in the southern Bay of Bengal authors have tried to document the bio-physical interactions, particularly for the evolution of surface/subsurface chlorophyll blooms, in this region during the summer monsoon. They have also used an OGCM to explain the dynamical processes relating to the nitrate limitations for the chl concentration. Considering the data sparsity in the Bay of Bangal, particularly for the biogeochemical data, this manuscript certainly contributes to enhance the existing literature of this region. However, I often find statements made

in this manuscript are not well supported by the figures. Below, I have listed some of them.

Also, I have serious doubt about the application of the model, particularly because spin-up time for the biogeochemical model is only 10 years, which is way too small for the nutrient levels to be stabilized. I believe, for such a basin scale TOPAZ, a minimum 30-50 years of spin-up is needed to stabilize the climatological nutrient levels, which will ultimately determine the surface chl concentration. Authors may plot the climatological simulation for the subsurface nitrate to see if that is stabilized. However, as this manuscript described the processes for a month long only and therefore, the results presented here might be unaffected by the slow drift in the nutrient levels of the model during the initial spin-up. But even then, a proper spin-up would be a good choice. Further, what about using open boundary conditions for the biogeochemical variables?

Page15, line 7-10: "The hydrodynamics of the region suggests that the triggering mechanism for bloom generation is open ocean Ekman pumping forced by positive wind stress curl Vinayachandran et al., 2004; Wijesekera et al., 2016a), favouring vertical transport of nutrients to the surface sunlit layers."

The authors relied too much on referencing. It is not difficult to calculate Ekman Pumping for the specific period. Authors are encouraged show that indeed the Ekman pumping is the primary driver. What about instability? This region exhibits one of the strongest barotropic/baroclinic instability of the north Indian Ocean.

Page 15, line no. 15: "The decay of surface bloom after 02 July (Fig. 5) followed the weakening of the dome (Fig. 3)." Not vary clear.

Page 15, lines 17-21: "CTD observations within the dome until 29 June, when the ship was at TSW, show that the subsurface chlorophyll concentrations were weak (< 0.5 mg m $-3$ ) just before the surface bloom event (Fig. 4e). This indicates that the vertical redistribution of subsurface phytoplankton does not have significant contribution in enhancing the surface chlorophyll. The generation of surface blooms is presumed to be dominantly controlled by the vertical transport of subsurface nutrients to the euphotic zone."

The mixed layer in SG579 does not seems shallowed considerably during the initial phase, but the chl concentration enhanced significantly in the mixed layer. The clear sky might be the major factor for this surface bloom as the authors said that the monsoon was active and therefore had considerable cloud cover in the previous week. It is possible that as the sky became clear it enhance the available light and thus marked by enhanced Chl. However, as the surface nutrients get consumed in few days the Chl concentration decreases again in spite of the persistence clear sky. How, authors can discard this possibility?

Page 16, lines 13-15: "Subsequent deepening of the mixed layer (âĹij70 m, Fig. 4d) suggests the role of mixing and entrainment in triggering the surface blooms."

What happens after 7th July when the MLD shallowed again in spite of increased wind speed? This does not explain authors hypothesis that the MLD deepens due to increased winds.

"The decay period of the bloom (08–10 July) coincided with the development of a freshening event. Surface salinity decreased by about 0.8 psu from 06 July to 10 July (Fig. 6) and the corresponding decrease in surface chlorophyll was about 0.27 mg m $-3$ (Fig. 5)."

The decrease of salinity during 6-10 July is of same order as seen during 4-5 July. This is only due to the fact that MLD shallows again and thereby inhibits the subsurface mixing of salinity. It may not be linked with lateral advection of fresh water and more to do with dynamics behind deepening of MLD during 5-7 July, which is not quite explained by the authors.

Later, in Figure 7 authors nicely explained the formation of barrier layer which inhibits

surface Chl. However, yet to convincingly explain why MLD deepens during 6-7 July. It may also help to extend the Figure 7 from 3rd July to see the barrier layer evolution.

Page 29, line 11: " Hence NO 3 was preferred over PO 4 and Fe (SiO4 does not limit growth in TOPAZ)"

I think this statement is not true. In TOPAZv1 large phytoplankton limitation term is dependent on Silicate. Please verify.

Figure 14: This figure is very confusing. It would help to overlay the weekly mean currents over the tendency terms. Many a times statements are made on vortices, SMC and its consequences on the NO3 budget, but without showing the mean currents it is very difficult to follow as a reader.

For example, authors said "Along the path of SMC, a clear patch of increased nitrate levels was evident (Fig. 14i), which extended from the southern tip of India up to about 85E. This indicates horizontal advection of coastally upwelled nutrients from the southern coasts of India and Sri Lanka (Fig. 14k) into the southern BoB by the SMC"

To me the NO3 show a negative tendency in the core of the SMC (east of SriLanka) and the positive patch may be along the edges. However, I can not make a concrete conclusions without any information of currents.

Further, authors claimed that Ekman pumping is the primary mechanism of surface Chl bloom, which I think is not well supported. Also, what about entrainment? At least SG620 show a clear signature of entrainment during 6-7 July.

Finally, what will be the effect of Rossby wave radiations from the eastern boundary of the Bay Bengal. Since, 8N is very close to the equatorial region, Rossby waves can travel pretty fast (∼20-25 cm/s ?) which means a Rossby wave front can cover about 2 degrees during the observation period and therefore, can implicate the east-west contrast between TSW and TSE.

---

## Referee Comment (RC2) · E. Boss (Referee) · 3 Oct 2018

Reviewer: Emmanuel Boss, University of Maine.

This paper is concerned with the dynamics of chlorophyll concentration in the Bay of Bengal, and uses observations from glider, a ship, satellite and a numerical model to describe it and attempt to understand it.

The paper's English is good. However, the English used is often not clear when it comes to the description of phytoplankton and their evolution. Bloom is never defined, sometime it seems to mean a relatively elevated chlorophyll concentration while in

other time it denote a positive change in time.

The paper may be of interest to readers of BG but I feel that it could be of significant more value if the authors addressed the following. I am also returning an annotated PDF (I stopped towards the end due to exasperation. Sorry.). I think addressing these will do a lot to make this paper significantly more useful.

1. The author adopt the classical view that phytoplankton dynamics are all determined by nutrients and light with physics modulating their availability. This view is not consistent with the fact that phytoplankton do not double in concentration daily even though they, on average, divide daily in most of the oceans (see review in ARMS by Ed Laws). This bottom up view is understandable given the lack of measurements to constrain losses, but the author should be very careful in their interpretation of temporal dynamics. In fact, in the height of the bloom, the maximal concentration, is when loss = growth. The recent paper by Behrenfeld and Boss, 2017, may make this point of view clearer to the writers.

Yes, productivity=growth rate x biomass, and hence when there is more chlorophyll there is likely more productivity.

2. The issue of photoaclimation is very important in stratified waters as chl/C can vary by factors as high as 5. The fact that the glider measure bbp as well as chlorophyll could be use to study this question. Similarly, the model you use should have variable Chl/C, unless you use bbp to estimate C_phyto (e.g. Graff et al., 2015)

3. If Fcdom is available (not clear what the 3rd channel of the triplet is) it could also be useful to understand light availability to phytoplankton.

4. Chlorophyll is a limited descriptor of biology (we don't know the species and the associated ecosystem from it). Limiting the text to describe its dynamics rather than talking about the 'biology' will make your text more palatable to some. In addition, the value you are estimating for it based on 'factory calibration' is likely biased by a factor

of 2 (e.g. Roesler et al., 2017).

5. The term 'bloom activity' is used over and over. What does that mean? Changes in chlorophyll concentrations? Try to be more precise.

6. Phytoplankton primary productivity is driven by PAR, which means they care about absorbing a photon in the visible but not about the energy of the photon (blue photons have about twice as much as red one). Your light model should be in PAR not W mˆ-2 and should take CDOM into account ('compete' with phytoplankton by absorbing blue photons).

Dear authors, I am often wrong. If you feel I am 'off the mark' feel free to contact me directly and if convinced, I will be more than happy to change my review.

Please also note the supplement to this comment:
https://www.biogeosciences-discuss.net/bg-2018-300/bg-2018-300-RC2-supplement.pdf

**Supplement:**

[revised manuscript text omitted]

---

## Author Comment (AC1) · 24 Oct 2018

**Reply to comments by Referee 1**

*(Referee's comments are given in blue and the response to comments are in black.)*

*Based on an observational campaign in the southern Bay of Bengal authors have tried to document the bio-physical interactions, particularly for the evolution of surface/subsurface chlorophyll blooms, in this region during the summer monsoon. They have also used an OGCM to explain the dynamical processes relating to the nitrate limitations for the chl concentration. Considering the data sparsity in the Bay of Bangal, particularly for the biogeochemical data, this manuscript certainly contributes to enhance the existing literature of this region. However, I often find statements made in this manuscript are not well supported by the figures. Below, I have listed some of them.*

The referee has made a constructive review on the manuscript which has helped to improve the analysis and presentation of results. We thank the referee for the comments and we have addressed each of them below.

*Also, I have serious doubt about the application of the model, particularly because spin-up time for the biogeochemical model is only 10 years, which is way too small for the nutrient levels to be stabilized. I believe, for such a basin scale TOPAZ, a minimum 30-50 years of spin-up is needed to stabilize the climatological nutrient levels, which will ultimately determine the surface chl concentration. Authors may plot the climatological simulation for the subsurface nitrate to see if that is stabilized. However, as this manuscript described the processes for a month long only and therefore, the results presented here might be unaffected by the slow drift in the nutrient levels of the model during the initial spin-up. But even then, a proper spin-up would be a good choice. Further, what about using open boundary conditions for the biogeochemical variables?*

Thanks to the referee for the suggestion. The time evolution of biological variables obtained from the ecosystem model shows that the spin up period of 10 years is fairly sufficient to address the processes that we are currtely looking at. Time series of chlorophyll and nitrate at different depth ranges in the southern Bay of Bengal from the

model spin up is shown in Fig. R1. Chlorophyll and nutrient levels show a stable annual cycle after 5-7 years of the spin up. Any model drift in the deep ocean circulation or nutrients is unlikely to affect the upper ocean bloom dynamics in the given time scale of interest. For the biogeochemical variables, no-flux condition has been applied at the open boundaries in the south and east. Open boundaries in the present model configuration are away from the study region and these boundaries have little impact on the model results for the timescale of our interest.

*Page15, line 7-10: "The hydrodynamics of the region suggests that the triggering mechanism for bloom generation is open ocean Ekman pumping forced by positive wind stress curl (Vinayachandran et al., 2004; Wijesekera et al., 2016a), favouring vertical transport of nutrients to the surface sunlit layers."*

*The authors relied too much on referencing. It is not difficult to calculate Ekman Pumping for the specific period. Authors are encouraged show that indeed the Ekman pumping is the primary driver. What about instability? This region exhibits one of the strongest barotropic/baroclinic instability of the north Indian Ocean.*

Thanks to the referee for pointing out this. We agree with the referee's comment on processes other than Ekman pumping in controlling the bloom dynamics in the region of the Sri Lanka Dome (SLD). Ekman pumping was calculated using ASCAT winds as shown in Fig. R2. Time series over the location of SG579 shows upwelling tendencies during most of the observational period (Fig. R2a). Pumping velocities peaked to about 2-3 m day$^{-1}$ by mid-June. Time series of minimum SLA shows that SLD attained its peak by the end of June (Fig. R2a), coinciding with the observed bloom at SG579. Ekman pumping remained to be upwelling favourable (0.4-0.7 m day$^{-1}$) during the period of surface bloom (30 June-02 July), though the magnitudes were relatively weaker. Strong upwelling in the second half of June, prior to the bloom event, is presumed to provide a favourable pre-conditioning by lifting the nitracline towards the surface. (This is explained below using the ecosystem model). Spatial distribution of mean Ekman pumping averaged for the BoBBLE observational period (24 June – 23 July) indicates widespread upwelling in the southern BoB (Fig. R2b).

During the decaying phase of the SLD in July, Ekman pumping velocities were positive, with peak values of about 2 m day$^{-1}$ (Fig. R2a). This indicates that the influence of remote

effects propagating from the eastern boundary of the BoB were dominant during this period (Vinayachandran and Yamagata 1998; Shankar et al., 2002; Wijesekera et al. 2016; Burns et al., 2017; Webber et al., 2018). Time-longitude hovmoller diagram of SLA from AVISO during May-July along 8°N, between 80-100°E is shown in Fig. R3. The decay period of SLD coincides with the arrival of positive SLA anomalies from east, representing the westward propagation of downwelling Rossby waves (Webber et al., 2018). This shows that, despite the positive Ekman pumping, remote forcings from the east contributed to the weakening of the SLD. The dynamics of the BoB is also characterised by instability effects associated with barotropic and baroclinic energy conversions (Vinayachandran and Yamagata, 1998; Kurien et al., 2010; Cheng et al., 2017). A complete energy analysis to examine the role of instability is beyond the scope of this paper. As far as the surface bloom generation is concerned, the proximity of nutricline to the surface (as well as the light availability, which will be explained in the following sections) is of primary concern. Hence we relied on the ecosystem model to identify the dominant forcing controlling the vertical displacement of nitracline.

Modelled SLD peaked on 28 June, two days prior to the observed peak.  The developing phase of simulated SLD (14-28 July) was characterised by the shoaling of nitracline. The shoaling rate of nitracline increased to about 1.0 m day$^{-1}$ by mid-June and closely followed the Ekman pumping velocities (Fig. R4). This shows that the vertical supply of nutrients to the surface layers during the developing phase of the SLD can be largely attributed to Ekman pumping. During the peak phase of SLD, both Ekman pumping velocities and nitracline shoaling rates weakened. However, the larger shoaling rates during the preceeding week indicate a favourable pre-conditioning for bloom generation during the peak phase of SLD. Later, during the decaying phase of the SLD in July, Ekman pumping gradually increased. However, the corresponding nitracline variability (with deepening tendencies) was not consistent with the pumping velocities, indicating the effect of remote forcings. These additional explanations will be added in Section 3.2.1 of the manuscript and Figures R2, R3 and R4 will be included.

*Page 15, line no. 15: "The decay of surface bloom after 02 July (Fig. 5) followed the weakening of the dome (Fig. 3)." Not vary clear*

Time series of minimum SLA in the region of the dome is shown in Fig. R2a. Sea level

anomalies decreased to about -0.3 m on 30 June. Intensification of SLD coincided with the observed surface bloom at SG579. The dome weakened afterwards as indicated by the weakening of negative SLA. The bloom decay after 02 July followed the weakening of the dome. This will be clarified in Section 3.2.1 of the manuscript and Fig. R2 will be included.

*Page 15, lines 17-21: "CTD observations within the dome until 29 June, when the ship was at TSW, show that the subsurface chlorophyll concentrations were weak (< 0.5 mg m$^{-3}$) just before the surface bloom event (Fig. 4e). This indicates that the vertical redistribution of subsurface phytoplankton does not have significant contribution in enhancing the surface chlorophyll. The generation of surface blooms is presumed to be dominantly controlled by the vertical transport of subsurface nutrients to the euphotic zone."*

*The mixed layer in SG579 does not seems shallowed considerably during the initial phase, but the chl concentration enhanced significantly in the mixed layer. The clear sky might be the major factor for this surface bloom as the authors said that the monsoon was active and therefore had considerable cloud cover in the previous week. It is possible that as the sky became clear it enhance the available light and thus marked by enhanced Chl. However, as the surface nutrients get consumed in few days the Chl concentration decreases again in spite of the persistence clear sky. How, authors can discard this possibility?*

We thank the referee for pointing out this possibility. Photosynthetically available radiation (PAR) from MODIS/VIIRS merged product along 8° N from June to July is shown in Fig. R5. The study region was under the influence of an active phase of the monsoon until the third week of June. This indicates light limitation on phytoplankton growth during this period. The active phase was followed by a convectively suppressed phase by the last week of June, one week prior to the glider deployment. In the region of SLD, PAR levels increased from about 12 E m$^{-2}$ day$^{-1}$ on 22 June to 50 E m$^{-2}$ day$^{-1}$ on 26 June. This shows that the transition from active to supressed phase favoured enhanced light availability for bloom generation. The glider data sampling began on 30 June, after the commencement of the suppressed phase and coinciding with the peak phase of the SLD. This restricts the identification of the relative importance of light and nutrient limitations on the generation of blooms at SG579. It may be noted that high radiation levels persisted till

mid-July, however, the surface layers exhibited oligotrophic conditions after the bloom decay. Surface chlorophyll dropped to levels below 0.1 mg m$^{-3}$ after 02 July (Fig 4a in the manuscript). This implies nutrient depletion in the surface layers resulting from phytoplankton consumption during the bloom event. The above details will be included in Section 3.2.1 of the manuscript. The shoaling of MLD is not so evident in the glider data, probably because the sampling period of SG579 starts on 30 June, when the SLD was already at its peak (Fig. R2a).

*Page 16, lines 13-15: "Subsequent deepening of the mixed layer (~70 m, Fig. 4d) suggests the role of mixing and entrainment in triggering the surface blooms."*
*What happens after 7th July when the MLD shallowed again in spite of increased wind speed? This does not explain authors hypothesis that the MLD deepens due to increased winds.*

Mixed layer shoaling after 07 July, despite the increase in wind speed, can be attributed to surface freshening. Surface salinity decreased by about 0.8 psu from 07 July to 10 July, inducing strong near-surface stratification (Fig. 6 in the manuscript). The resultant shoaling of mixed layer and the barrier layer formation (Fig. 7 in the manuscript) reveal the dominant role of freshwater over wind forcing in controlling the near-surface stratification and hence the surface blooms.

Deepening of mixed layer on 06 July occured between two freshening events; the first during 04-05 July and the second during 07-10 July (Fig. 6 in the manuscript). Surface salinity stratification was relatively weaker in between these events, providing conditions favourable for wind induced mixing. Taking into account the dynamic nature of the region, any impact of lateral transport of salinity on mixed layer deepening cannot be ignored. The quantification of lateral transport, however, is not feasible without estimates of advection and hence is outside the scope of the paper.

*"The decay period of the bloom (08–10 July) coincided with the development of a freshening event. Surface salinity decreased by about 0.8 psu from 06 July to 10 July (Fig. 6) and the corresponding decrease in surface chlorophyll was about 0.27 mg m −3 (Fig. 5)."*

*The decrease of salinity during 6-10 July is of same order as seen during 4-5 July. This is*

*only due to the fact that MLD shallows again and thereby inhibits the subsurface mixing of salinity. It may not be linked with lateral advection of fresh water and more to do with dynamics behind deepening of MLD during 5-7 July, which is not quite explained by the authors.*

*Later, in Figure 7 authors nicely explained the formation of barrier layer which inhibits surface Chl. However, yet to convincingly explain why MLD deepens during 6-7 July. It may also help to extend the Figure 7 from 3rd July to see the barrier layer evolution.*

The decrease in surface salinity cannot be completely attributed to mixed layer shoaling and inhibition of subsurface mixing of salinity. In the absence of an external freshwater source, mixed layer shoaling will not cause any further freshening in the surface layers as observed in the glider data. Since there was no local precipitation, the dominant mechanism which leads to surface freshening is presumed to be lateral advection. Impact of surface freshening on MLD and the barrier layer formation is evident from the profiles before and after the freshening event (Fig. 7 in the manuscript). Inhibition of mixing will finally limit the availability of nutrients in the surface layers and hence the surface bloom decays. Note that the shoaling of mixed layer occurs slightly later at the CTD location (Fig. 4e) and the surface bloom persists for longer here.

Figure 7 has been modifed by including selected daily mean profiles starting from 03 July till 10 July from SG620 (Fig. R6a-e). The barrier layer formation due to surface freshening can be observed during both the freshening events (Vinayachandran et al., 2018). Initial drop in surface salinity during the freshening events were of the same order (~0.4 psu; 04 July and 07 July in Fig. 6 of the manuscript). However, the first event was relatively shorter (04-05 July) and the second event lasted for a longer time period (07-10 July). Inhibition of surface blooms and the intensification of the DCM in the presence of surface salinity stratification can be observed during both the freshening events (Fig. R6b and R6d-e). Vertical profiles  obtained from CTD at TSE  for the same period are given in Figure R6f-j. CTD data shows both the freshening events, the associated development of barrier layers, the resultant decline in surface chlorophyll and the intensification of DCM, consistent with the glider observations (Fig. R6g and R6i-j). Above details and the modified Fig. 7 will be included in the manuscript.

*Page 29, line 11: " Hence NO 3 was preferred over PO 4 and Fe (SiO4 does not limit*

*growth in TOPAZ)" I think this statement is not true. In TOPAZv1 large phytoplankton limitation term is dependent on Silicate. Please verify.*

Thanks to the reviewer for the correction. SiO4 limits the growth of large phytoplankton in the model, but not considered in the case of small phytoplankton and diazotrophs. The text will be modified accordingly.

*Figure 14: This figure is very confusing. It would help to overlay the weekly mean currents over the tendency terms. Many a times statements are made on vortices, SMC and its consequences on the NO3 budget, but without showing the mean currents it is very difficult to follow as a reader.*

*For example, authors said "Along the path of SMC, a clear patch of increased nitrate levels was evident (Fig. 14i), which extended from the southern tip of India up to about 85E. This indicates horizontal advection of coastally upwelled nutrients from the southern coasts of India and Sri Lanka (Fig. 14k) into the southern BoB by the SMC"*

*To me the NO3 show a negative tendency in the core of the SMC (east of SriLanka) and the positive patch may be along the edges. However, I can not make a concrete conclusions without any information of currents.*

Thanks to the reviewer for the suggestion. Weekly mean currents are overlayed over the tendency terms (Fig. R7).  Horizontal advection of coastally upwelled nutrients from the southern coasts of India and Sri Lanka can be seen along the path of SMC ( Fig. R7i and R7k).

*Further, authors claimed that Ekman pumping is the primary mechanism of surface Chl bloom, which I think is not well supported. Also, what about entrainment? At least SG620 show a clear signature of entrainment during 6-7 July.*

Observations from SG579, in the region of SLD, shows no significant deepening of mixed layer during the period of surface blooms (Fig. 4a in the manuscript). This indicates that entrainment was relatively weaker here. Ouside the region of SLD, at TSE, SG620 shows deepening of mixed layer by about 20-30 m during $3^{rd}$ and $6^{th}$ July. This indicates a significant contribution of entrainment in the vertical supply of nutrients and hence the

surface bloom formation. Quantification of entrainment, however, requires additional information on the vertical gradient of nutrients.

*Finally, what will be the effect of Rossby wave radiations from the eastern boundary of the Bay Bengal. Since, 8N is very close to the equatorial region, Rossby waves can travel pretty fast (~20-25 cm/s ?) which means a Rossby wave front can cover about 2 degrees during the observation period and therefore, can implicate the east-west contrast between TSW and TSE.*

We thank the referree for the suggestion. Rossby waves propagating from the eastern boundary of the BoB can influence the depth of thermocline (nitracline) and hence the bloom activity. The east-west contrast between TSW and TSE is largely dependent on the spatial extent and strength of SMC and SLD, which is attributed to the combined effect of local as well as remote forcings. Observations of currents and sea level anomalies reveal that the location and intensity of SMC and SLD varied during the observational period (Fig. 3 in the manuscript). Using geostrophic velocities obtained from satellite data, Webber et al. (2018) showed that the SMC moved westward during the BoBBLE observational period. They related the westward shift of SMC to the westward propagation of downwelling Rossby waves from the eastern boundary of the BoB. The strength and spatial extend of SLD also varied accordingly. The decay period of the SLD coincided with the arrival of westward propagating high in sea level anomalies associated with the Rossby wave propagation (Fig. R3). The above explanations will be included in the manuscript.

**References**

Burns, J. M., B. Subrahmanyam, and V. S. N. Murty, ( 2017), On the dynamics of the Sri Lanka Dome in the Bay of Bengal, J. Geophys. Res. Oceans, 122, 7737–7750, doi:10.1002/2017JC012986.

Cheng, X., J. P. McCreary, B. Qiu, Y. Qi, and Y. Du, (2017), Intraseasonal-to-semiannual variability of sea-surface height in the astern, equatorial Indian Ocean and southern Bay of Bengal, J. Geophys. Res. Oceans, 122, doi:10.1002/2016JC012662.

Prescilla Kurien, Motoyoshi Ikeda and Vinu K. Valsala, (2010), Mesoscale Variability along the East Coast of India in Spring as Revealed from Satellite Data and OGCM Simulations, Journal of Oceanography, 66, 273–289

Shankar, D., Vinayachandran, P., and Unnikrishnan, A., (2002), The monsoon currents in the north Indian Ocean, Progress in Oceanography, 52, 63–120.

Vinayachandran, P. N. and Yamagata, T., (1998), Monsoon Response of the Sea around Sri Lanka: Generation of Thermal Domes and Anticyclonic Vortices, J. Phys. Oceanogr., 28, 1946–1960.

Vinayachandran, P. N., and Coauthors, (2018), BoBBLE (Bay of Bengal Boundary Layer Experiment): Ocean–atmosphere interaction and its impact on the South Asian monsoon. *Bull. Amer. Meteor. Soc.*,99, 1569–1587, https://doi.org/10.1175/BAMS-D-16-0230.1.

Webber, B. G. M., Matthews, A. J., Vinayachandran, P. N., Neema, C. P., Sanchez-Franks, A., Vijith, V., Amol, P., and Baranowski, D. B., (2018), The dynamics of the Southwest Monsoon current in 2016 from high-resolution in situ observations and models, Journal of Physical Oceanography, 48, 2259-2282.

Wijesekera, H. W., Shroyer, E., Tandon, A., Ravichandran, M., Sengupta, D., Jinadasa, S. U. P., Fernando, H. J. S., Agarwal, N., and Coauthors, (2016), "ASIRI: An Ocean–Atmosphere Initiative for Bay of Bengal", Bull. Am. Meteorol. Soc., 97, 1859–1884, https://doi.org/10.1175/BAMS-D-14-00197.1.

[Figure]

**Figure R1.** Time evolution of simulated chlorophyll (left panels) and nitrate (right panels) at the surface (top panels), 0-50 m (middle panels) and 50-100 m (bottom panels) for 10 years of the model spin up.

[Figure]

**Figure R2.** a) Time series of Ekman pumping (m day$^{-1}$; black) calculated from ASCAT winds around the location of SG579 (85-86° E, 7.5-8.5° N) and the minimum SLA (m; red) in the region of the Sri Lanka Dome (SLD) from 05 June to 20 July. b) Mean Ekman pumping averaged for the BoBBLE observational period (24 June – 23 July) in the southern BoB. Contours of SLA are overlayed.

[Figure]

**Figure R3.** Time-longitude hovmoller diagram of SLA (m) along 8°N between 81-100°E from May to July 2016.

[Figure]

**Figure R4.** Ekman pumping (m day$^{-1}$; black) and tendencies of nitracline (m day$^{-1}$; red), D26 (m day$^{-1}$; green) and D23 (m day$^{-1}$; magenta) averaged over the region of the modelled Sri Lanka Dome. Note that the tendency terms are reversed in sign so that positive (negative) values indicate shoaling (deepening). Nitracline is defined as the depth of 2 μmol kg$^{-1}$ nitrate isoline. Minimum sea level anomaly (m; blue) in the region of SLD is overlayed.

[Figure]

**Figure R5:** Photosynthetically available radiation (PAR; E m$^{-2}$ day$^{-1}$) along 8° N, between 83-92° E from June-July, 2016. The glider tracks during the BoBBLE field program are overlayed.

[Figure]

**Figure R6.** Daily mean vertical profiles of temperature (°C; red), salinity (psu; blue) and chlorophyll (mg m$^{-3}$; green) from (a-e) SG620 and (f-j) CTD at TSE location for selected days. The blue dashed line indicates the mixed layer depth, which is calculated as the depth where density is equal to the sea surface density plus an increase in density equivalent to 0.8°C. The red dashed line indicates isothermal layer depth (ILD) which is calculated as the depth where the temperature is cooler than SST by 0.8° C. The region between the MLD and ILD represents the barrier layer.

[Figure]

**Figure R7.** Model nitrate budget averaged over the mixed layer. Nitrate tendency (first column), vertical processes (second column), horizontal advection (third column) and the biological processes (fourth column) in μmol day$^{-1}$ are shown for 7-day averages starting from 24 June to 21 July 2016, marked on the left side of the corresponding panels. Vertical processes include vertical advection and mixing, and biological processes include source (nitrification) and sink (denitrification and uptake by the phytoplankton) terms for the model nitrate. Surface current (ms$^{-1}$) vectors are overlayed.

---

## Author Comment (AC2) · 24 Oct 2018

Please find our replies to comments by Dr. E. Boss.

We thank the referee for carefully reading the manuscript and offering the comments and suggestions. We have addressed each of them below.

Q. This paper is concerned with the dynamics of chlorophyll concentration in the Bay of Bengal, and uses observations from glider, a ship, satellite and a numerical model to describe it and attempt to understand it. The paper's English is good. However, the English used is often not clear when it comes to the description of phytoplankton and

their evolution. Bloom is never defined, sometime it seems to mean a relatively elevated chlorophyll concentration while in other time it denote a positive change in time. The paper may be of interest to readers of BG but I feel that it could be of significant more value if the authors addressed the following. I am also returning an annotated PDF (I stopped towards the end due to exasperation. Sorry.). I think addressing these will do a lot to make this paper significantly more useful.

A. We thank Dr. E. Boss for carefully reading the manuscript and offering the comments and suggestions. The term 'bloom' was used to refer to a condition of elevated chlorophyll concentration which is clearly distinguishable in space and time. During the study period, satellite observations of ocean color showed patches of high chlorophyll (0.3-0.7 mg m-3) in the regions of the Sri Lanka Dome (SLD) and the summer monsoon current (SMC), whereas the surrounding regions, outside the influence of these features, exhibited lower surface chlorophyll levels (< 0.2 mg m-3). These details will be included in the revised manuscript. The term bloom will be replaced by increase in chlorophyll appropriately.

Q. The author adopt the classical view that phytoplankton dynamics are all determined by nutrients and light with physics modulating their availability. This view is not consistent with the fact that phytoplankton do not double in concentration daily even though they, on average, divide daily in most of the oceans (see review in ARMS by Ed Laws). This bottom up view is understandable given the lack of measurements to constrain losses, but the author should be very careful in their interpretation of temporal dynamics. In fact, in the height of the bloom, the maximal concentration, is when loss = growth. The recent paper by Behrenfeld and Boss, 2017, may make this point of view clearer to the writers. Yes, productivity=growth rate x biomass, and hence when there is more chlorophyll there is likely more productivity.

A. Yes, chlorophyll distribution is determined by both physical and biological processes. In the present study, our main objective is to document the physical controls on the chlorophyll distribution, associated with the monsoon dynamics. In the southern BoB,

high chlorophyll concentrations were observed in the regions of strong dynamics, including the SLD and the SMC, indicating that the distribution of chlorophyll is largely dependent on the upper ocean dynamics. The biological controls are equally important. However, lack of observations on loss terms including mortality and sinking rates, and grazing by different zooplankton groups restricts a detailed investigation on their relative importance with respect to the physical processes during different stages of the chlorophyll bloom evolution. These limitations will be mentioned in the revised manuscript.

Q. The issue of photoaclimation is very important in stratified waters as chl/C can vary by factors as high as 5. The fact that the glider measure bbp as well as chlorophyll could be use to study this question. Similarly, the model you use should have variable Chl/C, unless you use bbp to estimate C_phyto (e.g. Graff et al., 2015)

A. We agree that photoacclimation is important. Considering the variability in available light as a function of depth, the physiological adaptation of phytoplankton through photoacclimation has a significant control on the chlorophyll concentration. The observed vertical distribution of chlorophyll does not necessarily represent the phytoplankton biomass distribution, since the chlorophyll to carbon ratio (Chl/C) can vary. The relation between chlorophyll and bbp has been examined using observations from SG620 at the time series location (Figure 1). This shows a linear relationship between chlorophyll and bbp, which indicates that the Chl:C ratio did not vary much in the region during the observational period.

The ecosystem model incorporates the effect of photoacclimation on the simulated chlorophyll distribution (Dunne et al., 2010). The model calculates a variable Chl:C which is dependent on light availability following Geider et al. (1997).

Q. If Fcdom is available (not clear what the 3rd channel of the triplet is) it could also be useful to understand light availability to phytoplankton.

A. Fcdom is not available at the time series location.

Q. Chlorophyll is a limited descriptor of biology (we don't know the species and the associated ecosystem from it). Limiting the text to describe its dynamics rather than talking about the 'biology' will make your text more palatable to some. In addition, the value you are estimating for it based on 'factory calibration' is likely biased by a factor of 2 (e.g. Roesler et al., 2017).

A. Thanks to the referee for the suggestion. The text will be modified accordingly in the revised manuscript.

Q. The term 'bloom activity' is used over and over. What does that mean? Changes in chlorophyll concentrations? Try to be more precise.

A. The term bloom activity was used indicate elevated levels of chlorophyll that is clearly identifiable in space and evolving in time. Generally, the surface layers of the southern Bay of Bengal exhibit oligotrophic conditions with chlorophyll concentrations below 0.2 mg m-3. During the summer monsoon, patches of enhanced chlorophyll concentrations (0.3-0.7 mg m-3) are observed in the dynamically active regions of the SLD and the SMC. By the end of the summer monsoon, chlorophyll levels decrease considerably. For clarity, the term 'bloom' will be replaced by 'chlorophyll' appropriately in the revised manuscript.

Q. Phytoplankton primary productivity is driven by PAR, which means they care about absorbing a photon in the visible but not about the energy of the photon (blue photons have about twice as much as red one). Your light model should be in PAR not W mËĘ-2 and should take CDOM into account ('compete' with phytoplankton by absorbing blue photons).

A. We agree with the referee that PAR is the appropriate parameter to explain primary productivity. At the time series location (89E, 8N), in situ observations of the vertical distribution of PAR is not available. Spatial and temporal coverage of attenuation coefficients derived from satellite data are also insufficient during the study period. The light model presented in the analysis (Morel and Antoine, 1994 and Manizza et al., 2005)

[Figure]

uses surface irradiance as input, instead of PAR. Hence, we preferred using the observed irradiance from the shipboard measurements to calculate the light penetration. The model considers light partioning into infrared and visible bands. Attenuation coefficients in the visible range were calculated seperately for two averaged wavelength bands (red and blue/green) at each depth levels as functions of chlorophyll profiles obtained from the glider (SG620), following Morel (1988). The vertically varying attenuation coefficients in the visible band will take into account the self-shading effect caused by the presence of phytoplankton in modulating the penetrative radiation into the subsurface layers, thereby influencing the DCM distribution.

PAR (E m-2 s-1) was estimated from the calculated penetrative radiation using the conversion,

PAR(z) = I_vis(z) * 2.75e18 / 6.023e23,

where I_vis(z) is the penetrative radiation (W m-2) in the visible range at depth z calculated using the light model, 2.75e18 quanta s-1 W-1 is the conversion factor obtained from Morel and Smith (1974), and 6.023e23 quanta E-1 is the number of photons corresponding to one mole. The depth of euphotic zone (Zeu) was calculated as the depth at which light reduces to 1% of the surface PAR value. Considering the fact that phytoplankton sees the absolute light level and not the percentage (Banse, 2004), the depth of threshold isolume (Z0.415), taken as the depth where PAR is 0.415 E m-2 day-1 below which light is insufficient to support photosynthesis (Letelier et al., 2004; Boss and Behrenfeld, 2010), is also shown (Figure 2).

Both Zeu and Z0.415 decreased during days with elevated surface chlorophyll (06-07 July) and deepened during days with weaker surface chlorophyll levels. This shows enhanced (reduced) light availability in the subsurface layers during days with low (high) surface chlorophyll, consistent with our results. Figure 10 in the manuscript and the corresponding text in Section 3.2.3. will be modfied in the revised manuscript following the above calculations. The location of study region is away from the coastal ocean

indicating less turbidity and relatively lower concentrations of CDOM, except those associated with the phytoplankton blooms. Hence we believe that exclusion of CDOM effect in the present light model will not affect the results significantly.

———————————————————

[Figure]

**Fig. 1.**

[Figure]

**Fig. 2.**

**Supplement:**

[revised manuscript text omitted]

---

## Author Response (AR2)

(Referee's comments are given in blue and the response to comments are in black. )

I read the revised manuscript. It is extremely descriptive (describing in chronological order observed features). The authors do not take full advantage of the sensors on the glider (the backscattering is not analyzed) nor are they taking full advantage of the model (which can provide them with model-consistent explanation to phenomena). The author can easily address photo-acclimation by using the chl/bbp ratio (after appropriate 'background' is removed from bbp). Bbp will also help in the interpretation of mixing events and depth integrated inventories.

We thank Dr. E. Boss for reading the revised manuscrpt and offering further comments and suggestions. We agree that the paper is descriptive, but this was necessary as the observed features had to be described in detail. We believe that the readers would find this useful. We do agree with Dr. Boss that extraction of additional information from both observations and model simulations are possible. However, considering the size of the manuscript, this is beyond the scope of this paper.

Backscattering was found to be generally higher in regions of deep chlorophyll maxima (DCM), located at a depth range of 30-60 m (Fig. R21). This indicates that the observed DCM can be associated with the biomass maxima and hence, cannot be completely attributed to photoacclimation. The glider chlorophyll and bbp exhibited a linear relationship (Fig. R1 in the Reply to first revision), which indicates that the Chl:C ratio did not vary much in the region during the observational period.

We have already included a nutrient budget analysis to present model consistent explanation for the mixed layer processes associated with the monsoon dynamics of the southern BoB which control the chlorophyll distribution. Further detailed analysis will be proceeded in future studies.

The paper is still full of 'bottom-up' statements such as: "However, the subsurface layers

still possess enough nutrients to support phytoplankton growth" – it is self-evident that where you find [chl] near the surface there are sufficient nutrient and light, even when [chl] are very low. As Laws (2105) suggest, even in what some considered to be oceanic 'deserts' cell divide daily meaning they have sufficient nutrients and light. Try to be more nuanced in your attempt to explain the observed.

We thank the reviewer for the suggestion. The sentence has been removed in the revised manuscript.

One way to think about the ocean plankton is as being very close to steady state with perturbation enhancing/reducing phytoplankton growth-rates. When growth-rates are enhanced, phytoplankton accumulation occurs, observed as an increase in biomass. When growth-rates are decreased, loss processes have the upper hand and biomass decreases. It is almost always result of small differences between loss and growth processes. The model you are using (within its assumptions) can help you diagnose this. A physical phenomena is often the cause of such small decoupling. I do not want to you to espouse a view that you do not believe in, but want you to think about the difference between accumulation rate (which you observe) and growth-rate (which you can estimate from a productivity model – but cannot calculate from your observations directly).

Thanks very much for this suggestion. The model can be used to examine the relative contribution of growth and loss terms during different phases of the phytoplankton evolution. Since the present manuscript is mainly intended to address the physical controls on the observed chlorophyll distribution, an in depth analysis on the simulated biological terms was not included. This will be proceeded in detail in future studies. The term phytoplankton growth has been replaced with phytoplankton accumulation appropriately.

Some other comments:
Your MLD criterion is clearly not a good mixing indicator (chlorophyll is stratified within it in some cases). Why not use a more stringent one?

The chlorophyll concentration is not always found to be homogeneous within the mixed layer, that is defined based on density. For example, the glider at TSE (SG620) shows a weak subsurface chlorophyll maxima within the mixed layer during the mixing events (06-08 July). Vertical profiles of chlorophyll, temperature, salinity and density for selected days

(05 July when the salinity stratification was strong and 06 July when there was strong mixing) are shown in Fig. R22. When the surface layers were stratified (05 July), the mixed layer was shallow and the DCM was located below the mixed layer. Within the mixed layer all the properties were near homogeneous. During the mixing event, the physical properties were well mixed within the mixed layer, which shows that the selected mixed layer criterion is reasonably good. However, chlorophyll was not found to be well mixed, probably due to the physiological adaptations of the phytoplankton to different environmental conditions. Therefore, we feel that this is not related to the MLD criterion.

The NPQ correction scheme does not work well as we still observe a strong diel periodicity in the chlorophyll time/depth series. If you see diel periodicity in bbp this will indicate that this periodicity is likely due to cell cycle and not NPQ.

The bbp data exhibits diurnal variability, consistent with chlorophyll fluorescence (Fig. R21). This shows that the NPQ correction is fairly good.

Provide uncertainties in the chlorophyll values you report (since you did not seem to calibrate them against environmental samples). Have you attempted to match them up with remote sensing at the surface?

These are the things we wish we could do better, but due to the inherent difficulties of glider data in remote regions with no simultaneous bottle sampling, POC measurements and HPLC calibrations, we are limited to what we have provided, and make our statements with the understanding that NPQ correction may not be perfect and the diel variability may not be fully resolved. A detailed comparision of glider data with satellite observations was not possible due to gaps in ocean color retrieval in the presence of cloud cover.

Are you using the exact same model parameters as those of Dunne (there are many to tune with)? If yes, say so. If not, provide those you use.

A table listing important biological parameters in the ecosystem model is added in the manuscript (Table 1). These are default values from the model source code and some are different from the values given in Dunne et al. (2010).

I suggested minor revisions as well as not to re-review a revised manuscript as I don't

want to impose my views on the authors. They should be free to publish the paper they want to publish.

Dear authors: the purpose of my review is to help you get the paper out there. If you feel my comments are 'off the mark' feel free to contact me and if convinced I will be more than happy to change them.

[Figure]

**Figure R21**. Optical backscatter (700 nm; m$^{-1}$) obtained from gliders a) SG579, b) SG534, c) SG532, and d) SG620.

[Figure]

**Figure R22**. Vertical profiles of chlorophyll (mg m$^{-3}$), temperature ($^{o}$C), salinity (psu) and density (kg m$^{-3}$) for 05 July 2016 (a-d respectively)  and 06 July 2016 (e-h respectively). The black horizontal line indicates the mixed layer 
[revised manuscript text omitted]